# X-linked deletion of *Crossfirre, Firre,* and *Dxz4* in vivo uncovers diverse phenotypes and combinatorial effects on autosomes

Tim P. Hasenbein [1,2,19], Sarah Hoelzl [1,2,19], Zachary D. Smith [3,4,5], Chiara Gerhardinger [3], Marion O. C. Gonner[1,2], Antonio Aguilar-Pimentel[6], Oana V. Amarie [6], Lore Becker [6], Julia Calzada-Wack[6], Nathalia R. V. Dragano[6], Patricia da Silva-Buttkus[6], Lillian Garrett[6,7], Sabine M. Hölter [6,7,8], Markus Kraiger [6], Manuela A. Östereicher[9], Birgit Rathkolb [6,10,11], Adrián Sanz-Moreno [6], Nadine Spielmann[6], Wolfgang Wurst [7,12,13], Valerie Gailus-Durner [6], Helmut Fuchs[7], Martin Hrabě de Angelis[6,10,14], Alexander Meissner [3,4,15,16], Stefan Engelhardt [1,2], John L. Rinn [17,18] ✉ & Daniel Andergassen [1,2,3] ✉

The lncRNA *Crossfirre* was identified as an imprinted X-linked gene, and is transcribed antisense to the *trans*-acting lncRNA *Firre*. The *Firre* locus forms an inactive-X-specific interaction with *Dxz4*, both loci providing the platform for the largest conserved chromatin structures. Here, we characterize the epigenetic profile of these loci, revealing them as the most female-specific accessible regions genome-wide. To address their in vivo role, we perform one of the largest X-linked knockout studies by deleting *Crossfirre, Firre,* and *Dxz4* individually and in combination. Despite their distinct epigenetic features observed on the X chromosome, our allele-specific analysis uncovers these loci as dispensable for imprinted and random X chromosome inactivation. However, we provide evidence that *Crossfirre* affects autosomal gene regulation but only in combination with *Firre*. To shed light on the functional role of these sex-specific loci, we perform an extensive standardized phenotyping pipeline and uncover diverse knockout and sex-specific phenotypes. Collectively, our study provides the foundation for exploring the intricate interplay of conserved X-linked loci in vivo.

During X chromosome inactivation (XCI), one female X chromosome becomes epigenetically silenced to compensate for the gene dose between males and females. In mice, XCI occurs in two successive waves, referred to as imprinted and random XCI. Imprinted XCI is initiated early in development by selectively inactivating the paternal X chromosome[1,2]. The paternal X chromosome remains silent in extraembryonic lineages[3] but undergoes reactivation during embryo implantation, followed by random XCI[4]. After the inactivation process, the inactive X chromosome (Xi) is folded in a compact chromatin structure known as the Barr body[5].

Chromosome conformation capture-based methods identified that the Xi folds into conserved megastructures and long-range interactions directed by the non-coding loci *Firre* and *Dxz4*[6-8]. Deleting *Dxz4* results in the loss of both megadomain and superloop formations[9-12]. In contrast, deleting *Firre* disrupts the superloop but does not affect the megadomains[11,13]. Studies in cell lines found that deletion of these loci has minimal effect on XCI biology beyond loss of

chromatin structures[9–12]. We previously addressed their in vivo role by generating mice lacking *Firre* and/or *Dxz4* and found that these loci are dispensable for mouse development and imprinted XCI biology[14]. At the same time, organ-specific expression changes were observed on autosomes, with the *Firre* locus identified as the main driver of these transcriptional changes[14].

Apart from the distinctive chromatin structures, allele-specific ChIP-seq experiments revealed that the *Firre* and *Dxz4* gene bodies contain a number of Xi-specific uncharacterized active transcription start sites and CTCF binding, which are thus present only in females[15,16]. Furthermore, both loci are transcribed into long non-coding RNAs (lncRNAs). While the *Dxz4* transcript (*4933407K13Rik*) is entirely uncharacterized, the lncRNA *Firre* was shown to play a role in multiple biological processes, including adipogenesis[17] and nuclear architecture[18]. Notably, *Firre* escapes random XCI in a distinctive manner, resulting in a full-length transcript from the active X chromosome (Xa) and multiple short isoforms from Xi[15,16,18]. A recent study provided compelling evidence that the *Firre* RNA controls autosomal genes in *trans*, by rescuing hematopoiesis defects in *Firre* knockout mice through the expression of a *Firre* transgene[19].

An additional X-linked lncRNA, *Crossfirre* (*Gm35612*), is transcribed antisense to the *Firre* lncRNA. *Crossfirre* consists of 3 exons and its 3′ end is located 500 bp from the 3′ end of *Firre*. In a comprehensive allele-specific analysis, *Crossfirre* was identified as an imprinted lncRNA in somatic tissues, predominantly transcribed from the maternal allele[15]. Since *Crossfirre* expression marks the maternal X chromosome, this locus may warrant further investigation for a link to imprinted XCI in addition to the maternal imprint controlling the *Xist* locus[20]. Moreover, *Crossfirre* is embedded in a 50 Kb repetitive cluster of long interspersed nuclear elements (LINE), DNA elements that are hypothesized to mediate the spreading of XCI into chromosomal regions that are otherwise prone to escape XCI[21,22].

To date, the in vivo role of *Crossfirre* in relation to imprinted and random XCI, both individually and in combination with *Firre* and *Dxz4*, has not been addressed. In addition, the role of *Firre* and *Dxz4* in random XCI biology has been difficult to study in vivo, as it requires an allele-specific single-cell analysis to overcome the random nature of XCI. Moreover, a detailed phenotypic and molecular characterization of mice lacking these loci is currently missing. Here, we first characterize the epigenetic profile of these loci and find them to be the most female-specific accessible regions genome-wide. To address their functional role in vivo, we generate one of the largest cohorts lacking non-coding X-linked loci by deleting *Crossfirre*, *Firre*, and *Dxz4* individually and in combination. Despite their distinct epigenetic features observed on the X chromosome, including the Xi-specific conserved megastructures and open chromatin as well as imprinted expression of *Crossfirre*, our extensive multi-omics investigation uncovers an interplay between *Crossfirre* and *Firre* in autosomal gene regulation, rather than affecting XCI biology. A subsequent phenotyping pipeline elucidates diverse phenotypes and sheds light on the functional role of these sex-specific loci. The resulting extensive molecular and phenotypic analysis provides the basis for further phenotypic and mechanistic characterization.

## Results

### *Crossfirre, Firre,* and *Dxz4* are the topmost female-specific chromatin accessible loci

Previously, an extensive allele-specific analysis identified the lncRNA *Crossfirre* as the only maternally expressed imprinted gene on the X chromosome[15]. *Crossfirre* is transcribed antisense to *Firre* and embedded in a 50 Kb repetitive LINE cluster (Fig. 1a, left). Imprinted expression of *Crossfirre* was observed in the brain and confirmed by H3K4me3 enrichment on the ERV-derived promoter in mouse embryonic fibroblasts (Supplementary Fig. 1a, b).

We first characterized the epigenetic profile of *Crossfirre, Firre* and *Dxz4* using published ATAC-seq data of the brain[23]. Our observations

confirmed the previously noted female-specific chromatin accessibility pattern within the *Firre* locus by revealing a more than twofold increase in the number of peaks in females ($n = 21$) compared to males ($n = 9$) (Fig. 1a left)[15,16,18]. An even more pronounced female-specific increase in chromatin accessibility was observed at the *Dxz4* locus (female peaks = 11, male peaks = 2), which forms contact with the *Firre* locus (Fig. 1a, right)[6–8]. To investigate whether this sex-specific pattern is common across tissues, we performed a genome-wide ATAC-seq analysis in a comprehensive set of organs[23] (Fig. 1b). This examination revealed *Crossfirre, Firre*, and *Dxz4* as the topmost female-specific regions in terms of chromatin accessibility across the genome (Fig. 1b, c; Supplementary Data 1, sheet c). Notably, the lncRNA *Xist* was identified as the 30th female-specific locus, emphasizing that the chromatin accessible pattern of *Crossfirre/Firre* and *Dxz4* can be used as additional markers for female tissues (Fig. 1c).

To determine whether this female-specific pattern derives from the Xa or Xi, we performed an allele-specific analysis on ATAC-seq data from clonal F1 neural progenitor cells[24]. In line with previous reports, we find that the female-specific chromatin accessibility in the *Crossfirre, Firre,* and *Dxz4* gene bodies is specific to Xi (Fig. 1d)[9,15,16]. In addition to the female-specific accessibility, the RNA of *Crossfirre, Firre,* and *Dxz4* is expressed at low to elevated levels in several adult mouse organs (Fig. 1e). Expression levels of *Crossfirre, Firre* and *Dxz4* in different tissues showed no significant correlation, indicating that these loci are not co-regulated (Supplementary Fig. 1c). Collectively, our findings demonstrate that *Crossfirre, Firre,* and *Dxz4* are the topmost female-specific accessible loci due to Xi-specific open chromatin.

### Mice carrying a *Crossfirre* single deletion or in combination with *Firre* and *Dxz4* are viable and undergo normal development

To investigate the in vivo role of these complex sex-specific loci, we generated various knockout mouse strains: deletion of the entire repetitive LINE cluster containing the *Crossfirre* gene alone (Δ*Crossfirre*), or in combination with the *Firre* locus (Δ*Crossfirre-Firre*), and a triple knockout (TKO) of *Crossfirre, Firre,* and *Dxz4* (Fig. 2a). Together with the previously published Δ*Firre* and Δ*Dxz4* single deletions, and the Δ*Firre-Dxz4* double deletion[14,19], this additional combination of knockout strains allows dissecting the contributions of the Xa-specific transcription of *Crossfirre, Firre,* and *Dxz4*, the LINE cluster and the Xi-specific open chromatin and megastructures in vivo.

Mutant founder mice were identified by PCR using primers spanning the deleted regions and subsequent Sanger sequencing of the PCR product (Supplementary Fig. 2a–c; Supplementary Data 1, sheet d). RNA-seq data of the spleen isolated from Δ*Crossfirre*, Δ*Crossfirre-Firre*, and TKO mice confirmed loss of expression in the deleted regions (Fig. 2b, c). The deletion of the *Crossfirre* locus did not affect the expression of the antisense transcript *Firre* (Fig. 2c). Similarly, the double deletion of *Crossfirre-Firre* did not affect the expression of the *Dxz4* transcript (Fig. 2c). Knockout mice of all three strains are viable and fertile and homozygous breeding results in the expected litter size and sex ratio, suggesting that these loci are dispensable for mouse development (Fig. 2d). These observations match the results of the previously published mice carrying the single and double deletion of *Firre* and *Dxz4*[14,19] and indicate that the addition of the imprinted *Crossfirre* locus does not play a vital role for mouse development.

### Deletion of the imprinted *Crossfirre* locus alone or in combination with *Firre* and *Dxz4* does not affect imprinted XCI

Given that *Crossfirre* is an imprinted gene expressed from the maternal X chromosome, we speculate that the locus may serve as a marker to prevent silencing of the maternal X chromosome in extraembryonic lineages with imprinted XCI. To address this question, we isolated E12.5 female placentas from reciprocal crosses between all the generated *Crossfirre* mutants and *Mus musculus castaneus* CAST/EiJ (CAST) and

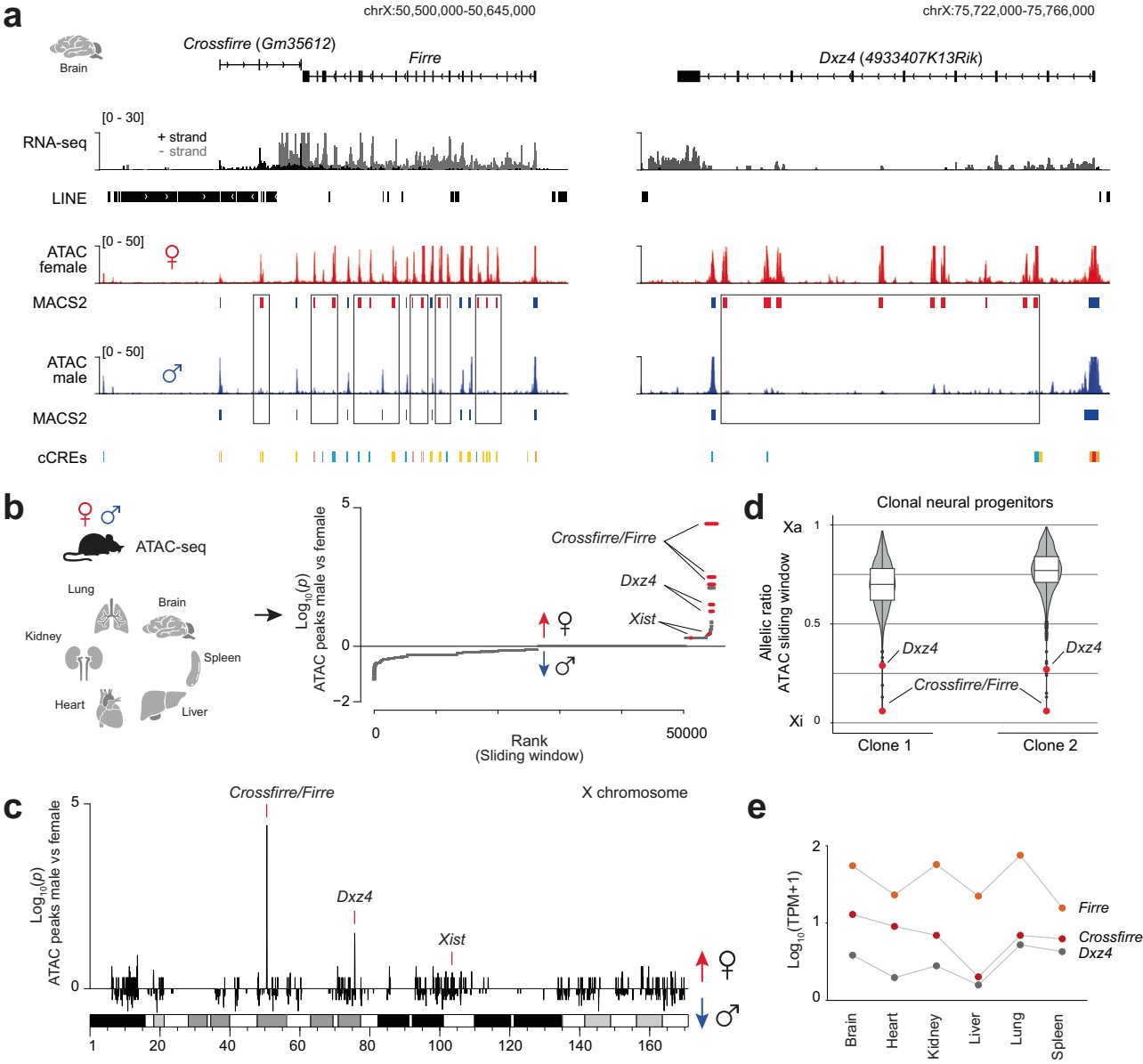

**Fig. 1 | *Crossfirre*, *Firre*, and *Dxz4* loci are the topmost female-specific loci by chromatin accessibility with unique allele-specific characteristics. a** Genome browser tracks of the mouse brain showing strand-specific RNA-seq[15] and ATAC-seq[23] for females (red) and males (blue) covering the *Crossfirre* (*Gm35612*), *Firre*, and *Dxz4* (*4933407K13Rik*) locus. The gene body of *Crossfirre* is embedded in a 50 Kb LINE cluster. Boxes highlight female-specific ATAC-seq peaks identified through MACS2[60]. ENCODE Candidate Cis-Regulator Elements (cCRES) are shown for each locus (Promoter: red, Proximal enhancer: orange, Distal enhancer: yellow, DNase/H3K4me3: pink, CTCF: blue). **b** Overview of six female and male organs (two biological replicates per sex, *n* = 24) utilized to map sex-specific chromatin accessible loci from publicly available ATAC-seq data[23] (left). A one-sided binomial test was used to identify sex-specific loci genome-wide by comparing male and female ATAC peak counts binned over 100 Kb sliding windows (right, see section

"Methods"). A positive value was assigned if more peaks were present in females than in males, while negative values were assigned if the reverse was true. The top female-specific loci *Crossfirre*, *Firre*, and *Dxz4* are indicated, together with the expected female-specific *Xist* locus. **c** Sex-specific analysis of (**b**) for the entire X chromosome. **d** Violin plots displaying the allelic ratio of ATAC enrichment over a 50 Kb sliding window for the entire X chromosome of neuronal progenitor cell clones (*n* = 2)[24]. The red dot highlights the allelic ratio of the sliding window overlapping the *Crossfirre*/*Firre* and *Dxz4* locus. Boxes represent the interquartile range around the median, while the whiskers extend to 1.5 times the interquartile range. **e** RNA-sequencing abundance of *Crossfirre* (red), *Firre* (orange), and *Dxz4* (gray) across various mouse organs. Shown are the log$_{10}$-transformed mean TPM values of each gene within the respective tissue with a pseudocount of 1 added.

performed RNA-seq (Fig. 3a). We first examined the expression levels of *Crossfirre*, *Firre*, and *Dxz4* in TKO. All three lncRNAs could not be detected in the maternal deletion (Fig. 3a, right). At the same time, wildtype (WT) levels were observed for the paternal deletion (Fig. 3a, left), implying that these lncRNAs are expressed exclusively from Xa in the placenta. Thus, this system allows us to disentangle the functional role of the lncRNAs on Xa from the female-specific epigenetic signatures that only exist on Xi.

We performed a differential gene expression analysis for Δ*Crossfirre*, Δ*Crossfirre-Firre*, and TKO mutants, as well as for our previously published placenta data, including Δ*Firre*, Δ*Dxz4*, and Δ*Firre-Dxz4* (FDR ≤ 0.01, |shrunk log2FC| ≥1)[14]. Independent of whether these loci are absent on Xa or Xi, individually or in combination, we detected very few dysregulated genes (Fig. 3b; Supplementary Data 1, sheet e). To validate the identified candidates, we compared the dysregulated genes between TKO and the double deletions Δ*Crossfirre-Firre* and

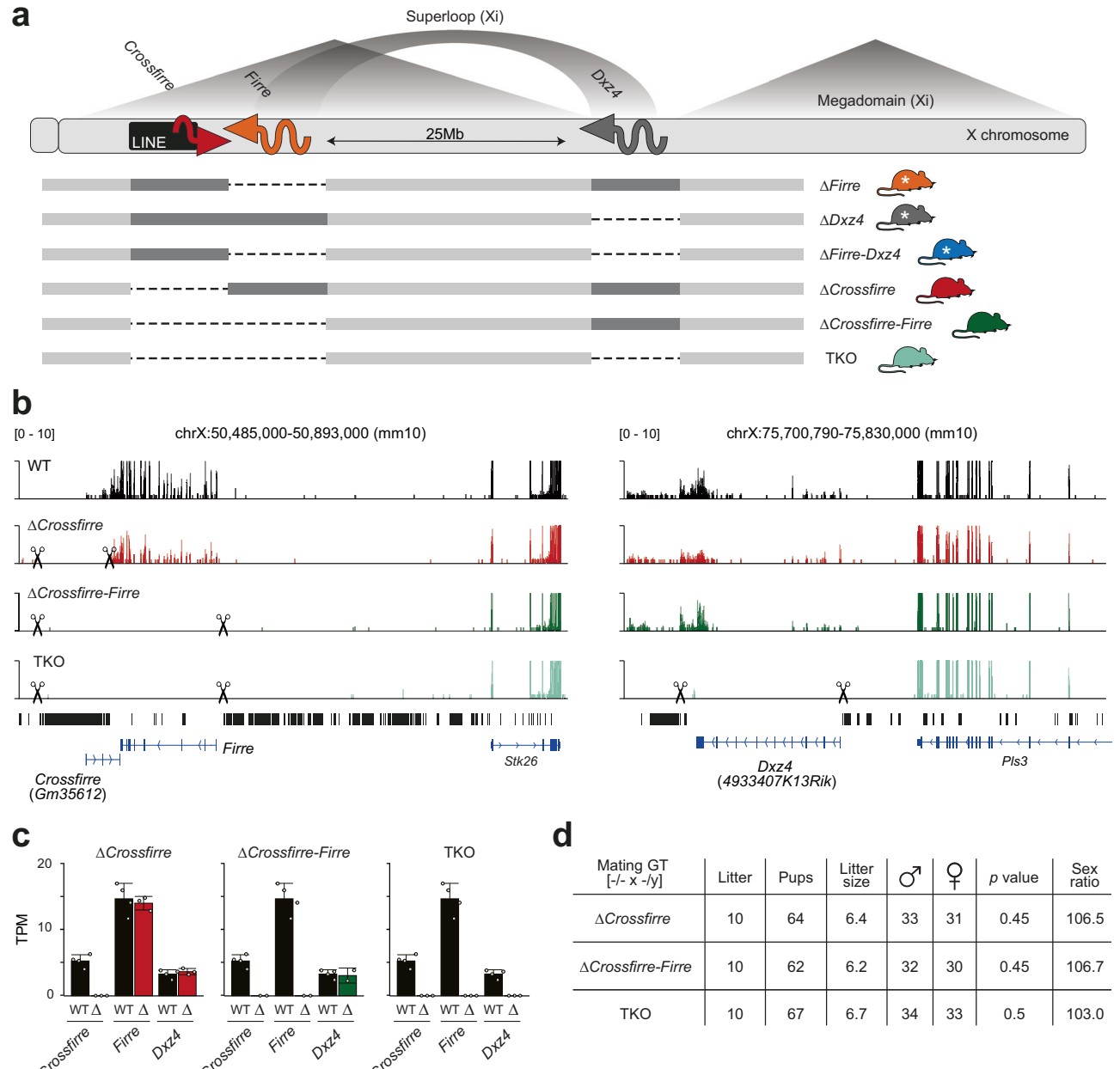

**Fig. 2 | Mice carrying a *Crossfirre* single deletion or combined with *Firre* and *Dxz4* are viable and undergo normal development. a** Schematic overview of the X chromosome. The megadomains and the superloop between the *Firre* and *Dxz4* loci are specific to the inactive X chromosome (Xi), whereas full-length transcription of *Firre* and *Dxz4* only occurs on the active X chromosome (Xa). The imprinted lncRNA *Crossfirre* shows expression from the maternal X chromosome, independent of random XCI[15]. Dotted lines indicate the deleted loci for ΔFirre (orange), ΔDxz4 (gray), ΔFirre-Dxz4 (blue), ΔCrossfirre (red), ΔCrossfirre-Firre (green), ΔCrossfirre-Firre-Dxz4 (TKO, turquoise). Colors are used in figures throughout the manuscript to highlight the genotype of origin. White stars refer to previously generated mouse strains[14]. **b** Genome browser RNA-seq tracks showing the *Crossfirre*/*Firre* (left) and *Dxz4* (right) locus for the adult spleen of wildtype (black), and ΔCrossfirre, ΔCrossfirre-Firre, TKO mutants. Scissors indicate the start and end of the CRISPR-Cas9 deletion (see section "Methods"). **c** Mean expression values of *Crossfirre*, *Firre*, and *Dxz4* in the adult spleens of *Crossfirre* mutant strains (WT n = 4, ΔCrossfirre n = 3, ΔCrossfirre-Firre n = 2, TKO n = 3). Error bars indicate the standard deviation. **d** Sex distribution of homozygous ΔCrossfirre, ΔCrossfirre-Firre, and TKO breeding. The *p* values are obtained from a one-sided binomial test.

ΔFirre-Dxz4 (Fig. 3c). We found that *Firre* was the only dysregulated gene shared in all strains with the deletion on Xa, while three pseudogenes were shared between the TKO and ΔCrossfirre-Firre (ΔXa: *Gm13340*, *Gm13436*, and ΔXi: *Rpsa-ps10*, Fig. 3c).

Subsequently, we conducted an allele-specific RNA-seq analysis to investigate if the absence of these loci results in deviations of the expected maternal allelic ratios for all the X-linked genes or in the local regions of *Crossfirre*, *Firre*, and *Dxz4*. Similar to previous observations, we identified strain-specific escape, with more genes escaping from CAST Xi compared to BL6 Xi[15,25,26] (Fig. 3d, e, Supplementary Fig. 3a, b).

We found that regardless of whether the deletions were on Xi or Xa, the median allelic ratios of informative X-linked genes were unchanged in all investigated knockout strains (Fig. 3d, e; Supplementary Data 1, sheet f). The *Crossfirre* knockout also incorporates the deletion of an entire LINE cluster on the X chromosome, DNA elements that have been suggested to prevent gene escape[21,22]. Thus, we investigated if deletion of these loci, including the LINE cluster, leads to alterations in the allelic ratios in the proximity and found that the absence of these loci has no effect on neighboring genes in *cis* (Fig. 3e, Supplementary Fig. 3c). In summary, we found that the lack of the imprinted *Crossfirre*

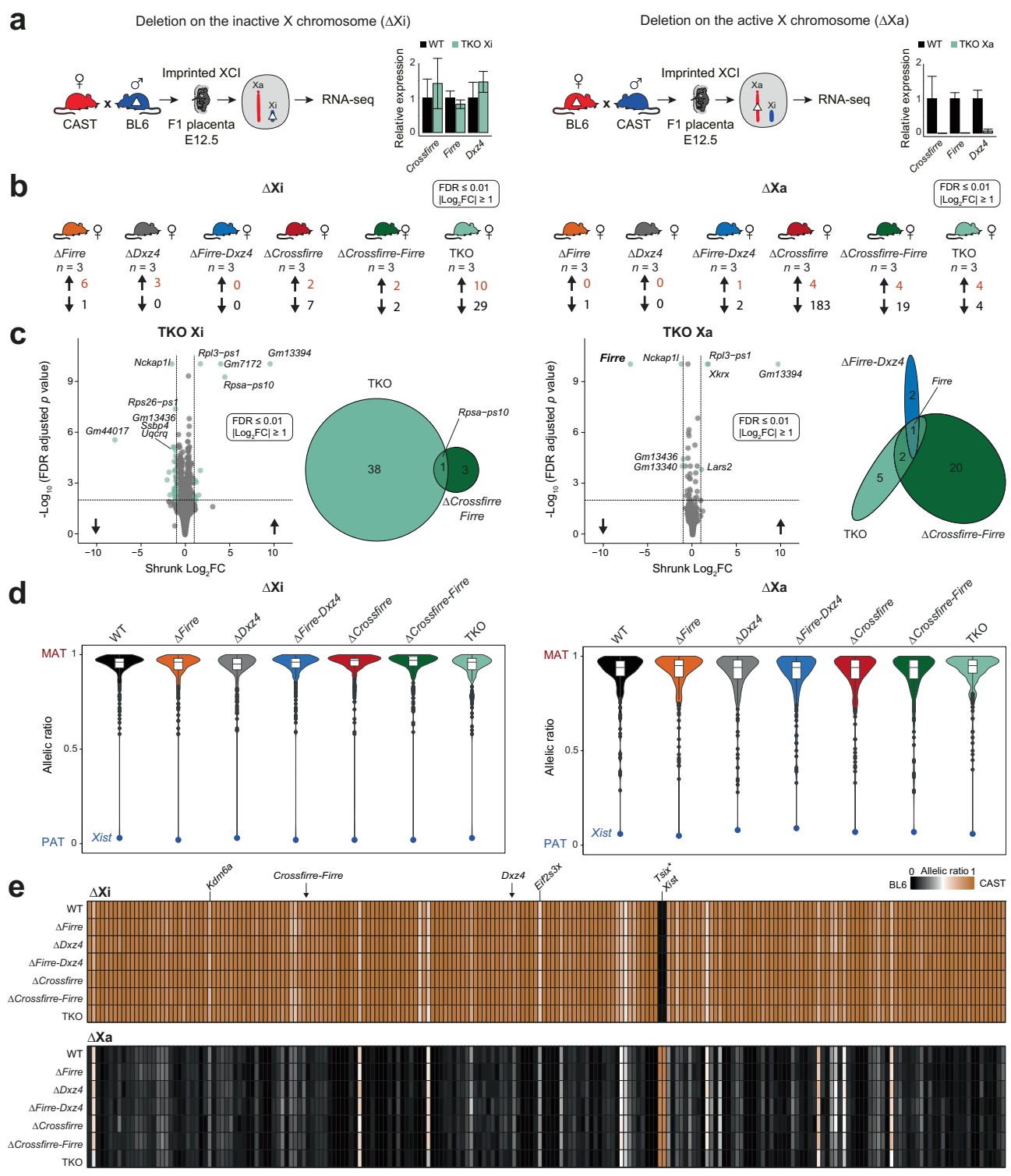

**a** Deletion on the inactive X chromosome (ΔXi) — Deletion on the active X chromosome (ΔXa)

locus alone or in combination with *Firre* and *Dxz4* has no impact on imprinted XCI in the placenta, regardless of whether the deletion is present on Xi or Xa.

### Deletion of topmost female-specific loci *Crossfirre*, *Firre*, and *Dxz4* does not affect random XCI

The precise influence of these loci on random XCI and escape in vivo was challenging to assess in previous studies due to the absence of single-cell analysis in mouse organs. Given the random nature of XCI, obtaining the transcriptome of a single-cell allows us to investigate

whether the TKO on the Xa or Xi has an impact on random XCI. Hence, we crossed heterozygous TKO females with CAST males, isolated adult F1 spleens from heterozygous females and WT littermates, and performed single-cell RNA-seq (Fig. 4a). After quality control and normalization, we obtained the allele-specific single-cell transcriptome for 2043 WT and 1642 heterozygous cells (see section "Methods"). Subsequent clustering analyses displayed the expected cell types for WT and TKO (Fig. 4b). Next, we performed an allele-specific expression analysis at the chromosome level by combining all X chromosomal reads to compute whether the BL6 or the CAST X

**Fig. 3 | Deleting the imprinted *Crossfirre* locus alone or together with *Firre* and *Dxz4* does not affect imprinted XCI. a** Schematic of our experimental system to investigate the impact of the deletions on the inactive X (Xi, left) or active X (Xa, right) for imprinted X inactivation. RNA-seq data of E12.5 female placentas are analyzed from wildtype (WT) F1 reciprocal crosses (CASTxBL6 $n = 9$, BL6xCAST $n = 8$), and for the six F1 mutants carrying the deletion on the paternal Xi ($n = 3$ per genotype) or on the maternal Xa ($n = 3$ per genotype). The relative expression (mean and standard deviation) between WT and Δ*Crossfirre-Firre-Dxz4* (TKO, turquoise) is shown for deletions on Xi (left) and Xa (right). **b** Number of differentially expressed genes across mutant strains (DEseq2: FDR ≤ 0.01, |log2FC| ≥ 1). **c** FDR-adjusted *P* values of differential gene expression analysis between WT and TKO on Xi (left) and Xa (right). Venn diagram showing the overlap of dysregulated genes between Δ*Firre-Dxz4* (blue), Δ*Crossfirre-Firre* (green), and the TKO (turquoise).

**d** Median allelic ratios for X-linked genes in WT (black) and the six knockout strains carrying the deletions on Xi (left) or Xa (right). The blue dot emphasizes the paternal allelic ratio of the lncRNA *Xist*. The allelic ratios range from 0 to 1 such that 1 corresponds to maternal expression (MAT), 0.5 to biallelic expression, and 0 to paternal expression (PAT). Boxes indicate the interquartile range around the median and whiskers 1.5x the interquartile range. **e** Heatmap showing median allelic ratios for X-linked genes that are informative across all WT and knockout strains carrying the deletions on Xi (upper panel) or Xa (lower panel). Brown indicates an allelic ratio of 1 (CAST allele), while black indicates an allelic ratio of 0 (BL6 allele). Common escape genes *Kdm6a* and *Eif2s3x* are highlighted showing biallelic expression, thus validating our approach. Arrows indicate the approximate location of *Crossfirre*, *Firre*, and *Dxz4*. *The expression of *Tsix* from Xi is due to the overlapping nature with *Xist* and thus an artifact of the non-stranded analysis.

chromosome is inactive (see section "Methods"), permitting us to divide the single-cell RNA-seq data into WT CAST Xa ($n = 1342$), WT BL6 Xa ($n = 640$), TKO on Xi (CAST Xa, $n = 1359$), and TKO on Xa (BL6 Xa, $n = 243$, Fig. 4c, Supplementary Fig. 4a, b). A small fraction of cells with two Xa were excluded, which may result from duplicates (WT = 61, TKO = 40, Supplementary Fig. 4b). We found the expected X inactivation skewing ratio in the WT spleen population (65.7% CAST Xa and 31.3% BL6 Xa), a known phenomenon in F1 crosses between BL6 and CAST strains resulting in the predominant inactivation of the BL6 X chromosome[27]. Notably, we detected a more pronounced skewing ratio in the TKO heterozygous spleen cell population (82.8% TKO on Xi and 14.8% TKO on Xa, Fig. 4c, Supplementary Fig. 4b, c). To verify if the heterozygous TKO has an impact on the skewing ratio, we repeated the experiment using bulk RNA-seq on WT ($n = 3$) and −/+ TKO ($n = 3$) spleens (Supplementary Fig. 4d). In contrast to the results of the single-cell data, we observed a comparable skewing ratio between WT and heterozygous TKO samples, with no significant differences observed for the *Xist* allelic ratio (Supplementary Fig. 4e, f).

To investigate the impact of the TKO on random XCI at gene level, we performed an allele-specific analysis on the four datasets: WT CAST Xa, WT BL6 Xa, TKO on Xi, and TKO on Xa. In order to increase gene coverage, we aggregated read counts as pseudobulk. Subsequently, we applied Allelome.PRO to calculate allelic ratios of X-linked genes for each condition, enabling a background-matched comparison between WT and TKO on Xi or Xa. Using this approach, we successfully identified *Xist* being exclusively expressed from Xi and known escape genes: *Kdm6a*, *Eif2s3x*, *Ftx*, and *Kdm5c*, validating our approach. Consistent with the result for imprinted XCI, we did not observe significant differences in the allelic ratios of X-linked genes between the WT and TKO, regardless of whether the deletion was present on Xi or Xa (Fig. 4d, Supplementary Data 1, sheet g). These results lead us to conclude that neither the Xi-specific chromatin structures and accessibility, nor the lncRNA product, and the 50 Kb LINE cluster have an impact on XCI biology in vivo.

### Deletion of X-linked loci *Crossfirre*, *Firre*, and *Dxz4* results in autosomal dysregulation

Given that *Firre* and *Dxz4* have previously been implicated in autosomal gene regulation[14,18,19,28], we investigated whether deletion of *Crossfirre* alone or in combination with *Firre* and *Dxz4* has functional roles outside of XCI biology. Thus, we generated a transcriptomic bodymap from homozygous TKO mice, including adult spleen, kidney, lung, heart, liver, and brain (Fig. 5a, Supplementary Fig. 5a). We further reanalyzed the previously published age- and organ-matched Δ*Firre-Dxz4* female bodymap, where we observed spleen-specific autosomal gene dysregulation[14]. Overall, we find a total of 1190 differentially expressed genes in the TKO and 104 in the Δ*Firre-Dxz4* (FDR ≤ 0.01, |shrunk log2FC| ≥ 1), implying that the addition of the *Crossfirre* deletion results in an 11.4x fold increase of dysregulated genes (Fig. 5a; Supplementary Data 1, sheet h−i). The highest number of TKO

differentially expressed genes was detected in the spleen ($n = 417$), whereof most were located on autosomes rather than the X chromosome (mean: 96.3% autosomes, 3.7% X chromosome, Fig. 5a). Of note, we did not observe a correlation between the WT expression levels of these loci and the number of differentially expressed genes (Supplementary Fig. 5b). Interestingly, 148 dysregulated genes were shared in two or more organs, and the majority matched the direction of dysregulation (93.2%, Fig. 5b, Supplementary Data 1, sheet j). The conservative set of dysregulated genes (differentially expressed in ≥5 organs) is upregulated across all organs. To identify dysregulated pathways in the TKO organs we performed a gene set enrichment analysis (GSEA, FDR ≤ 0.1) and found that the majority of dysregulated gene sets were upregulated and shared between organs with the brain as the exception (Fig. 5c left; Supplementary Fig. 5c; Supplementary Data 1, sheet k). Additional network analysis of the spleen revealed two dominant clusters associated with ribosomal (cluster ID 3, $n = 35$ of 100) and mitochondrial pathways (cluster ID 1, $n = 21$ of 100, Fig. 5c right). Taken together, these results suggest that the additional deletion of *Crossfirre* leads to a substantial increase in gene dysregulation on autosomes, shared across organs.

### *Crossfirre* affects autosomal gene regulation, but only in combination with *Firre*

To clarify whether *Crossfirre* alone or in combination with *Firre* and *Dxz4* is the main driver of the observed molecular phenotype in TKO, we examined the individual deletions in the spleen, the organ with the most robust phenotype. We performed differential gene expression analysis for homozygous Δ*Firre*, Δ*Dxz4*, Δ*Firre-Dxz4*, Δ*Crossfirre*, Δ*Crossfirre-Firre*, and TKO mutants using a conservative cutoff (FDR ≤ 0.01, |shrunk log2FC| ≥ 1, Fig. 5d, Supplementary Data 1, sheet h and l). The highest number of dysregulated genes was observed in mutants lacking *Crossfirre* and *Firre* in combination (TKO: $n = 417$, Δ*Crossfirre-Firre*: $n = 103$), with numerous overlapping genes sharing directionality ($n = 73$, 70.87%, Fig. 5e; Supplementary Data 1, sheet m). Remarkably, the single deletions alone result in a low number of dysregulated genes, suggesting a combined effect. Subsequent GSEA for each knockout strain revealed that only the combined *Crossfirre-Firre* deletion could reproduce the upregulated pathways observed in TKO mutants (Fig. 5f; Supplementary Data 1, sheet n). In summary, these results suggest that the simultaneous deletion of *Crossfirre* and *Firre* is the main driver of autosomal gene dysregulation observed in TKO mutants.

### Comprehensive phenotyping pipeline uncovers knockout- and sex-specific phenotypes

While the TKO mutants showed no obvious impairment in fertility, viability, and sex ratios, the results of our molecular analysis revealed widespread gene dysregulation in multiple organs. To further investigate the phenotypic characteristics of these loci, we subjected a TKO mouse cohort of both sexes to the standardized phenotyping screening pipeline of the German Mouse Clinic (GMC)[29]. Mice

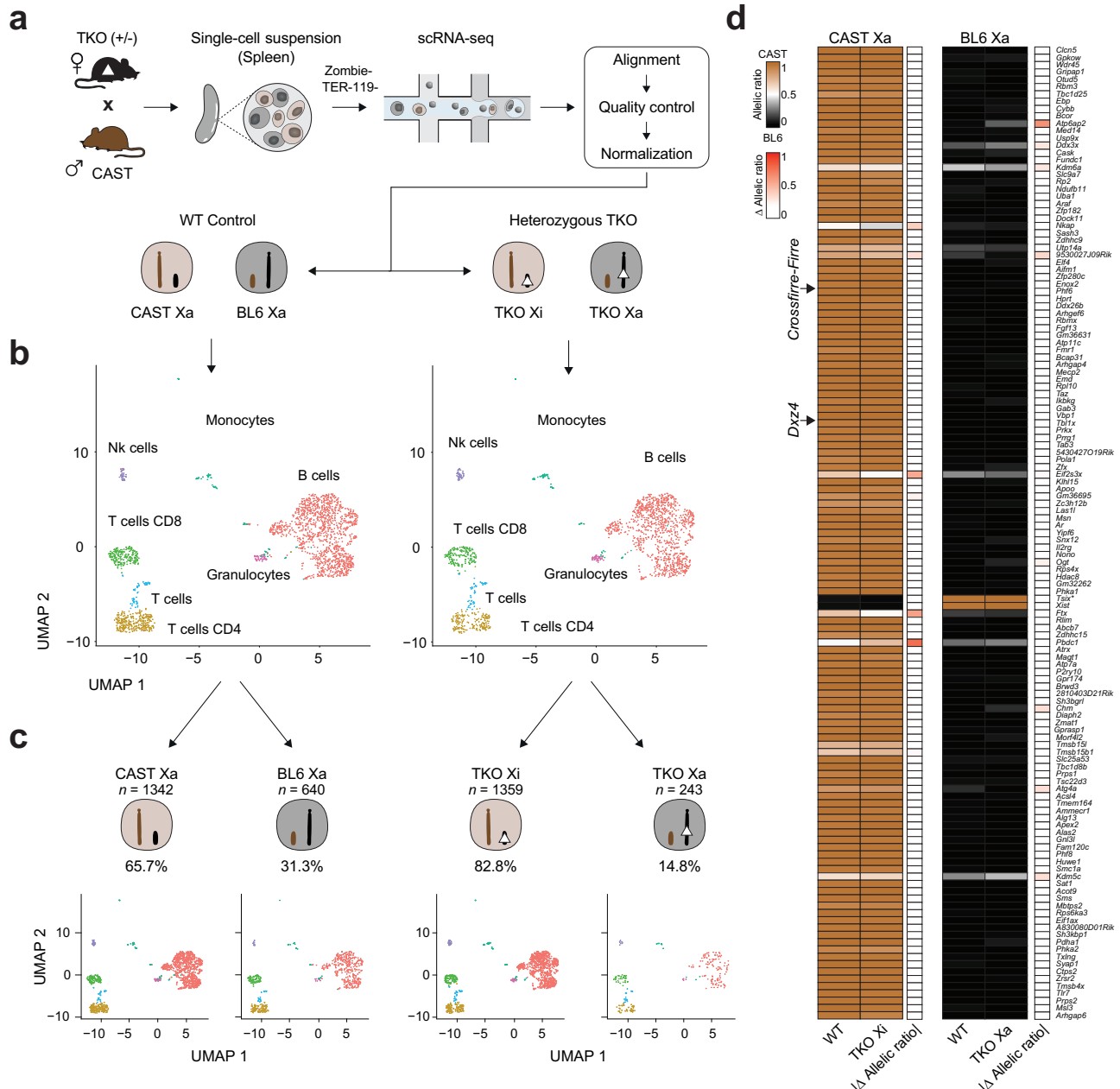

**Fig. 4 | Deletion of topmost female-specific loci *Crossfirre*, *Firre*, and *Dxz4* does not affect random XCI. a** Schematic overview of the experimental setup to investigate the effects of the Δ*Crossfirre-Firre-Dxz4* (TKO) on the active X (Xa) or inactive X (Xi) upon random XCI. Heterozygous TKO females (BL6) were mated with wildtype (WT) CAST mice to generate F1 hybrids with WT and heterozygous TKO genotypes. Spleens were isolated (WT *n* = 1, TKO heterozygous *n* = 1) and processed for single-cell RNA-seq. Sequencing data were subjected to bioinformatic preprocessing, including alignment, quality control, and normalization. The dataset was split according to the genotype condition (WT/TKO). **b** UMAP of unsupervised clustering by genotype condition (WT/TKO). **c** Single-cell RNA-seq data were further split by Xa chromosome state (CAST Xa, BL6 Xa) using

Allelome.PRO and a chromosome-wide window (see section "Methods"). Thus, we obtained single-cell RNA-seq data from WT and TKO spleen cells which could be categorized in one of four groups: WT CAST Xa, WT BL6 Xa, TKO Xi (CAST Xa), and TKO Xa (BL6 Xa). **d** Heatmap showing the median allelic ratios for informative X-linked genes in WT and TKO mice carrying the deletions on Xi (left) or Xa (right). Read counts of all cells were summarized as pseudobulk to increase gene coverage. The brown color indicates an allelic ratio of 1 corresponding to the CAST allele, while black indicates an allelic ratio of 0 (BL6 allele). Arrows indicate the approximate location of *Crossfirre*, *Firre*, and *Dxz4*. *The expression of *Tsix* from Xi is due to the overlapping nature with *Xist* and thus an artifact of the non-stranded analysis.

underwent diverse phenotyping modules including hundreds of measurements, to assess physiological parameters from the following categories: immunology/allergy, behavior, biomarkers, cardiovascular, clinical chemistry, pathology, dysmorphology, eyes, metabolism, neurology, and nociception (Fig. 6a). Based on the parameters tested we found a total of 28 significant knockout- and sex-specific phenotypes, affecting immunology/allergy (*n* = 5), behavior (*n* = 2), neurology (*n* = 1), cardiovascular (*n* = 2), clinical chemistry (*n* = 8),

dysmorphology (*n* = 2), metabolism (*n* = 3), and pathology (*n* = 5, Fig. 6b, c and Supplementary Data 1, sheet o–q).

Regardless of sex, we identified nine phenotypes that were specific to TKO. These included increased locomotor and exploratory activity and decreased acoustic startle reactivity. Moreover, mutant mice exhibited alterations in red blood cell morphology, mild effects on iron metabolism, and increased secondary follicles in the Peyer's patches of the intestine (Fig. 6b, c and Supplementary Data 1, sheet o).

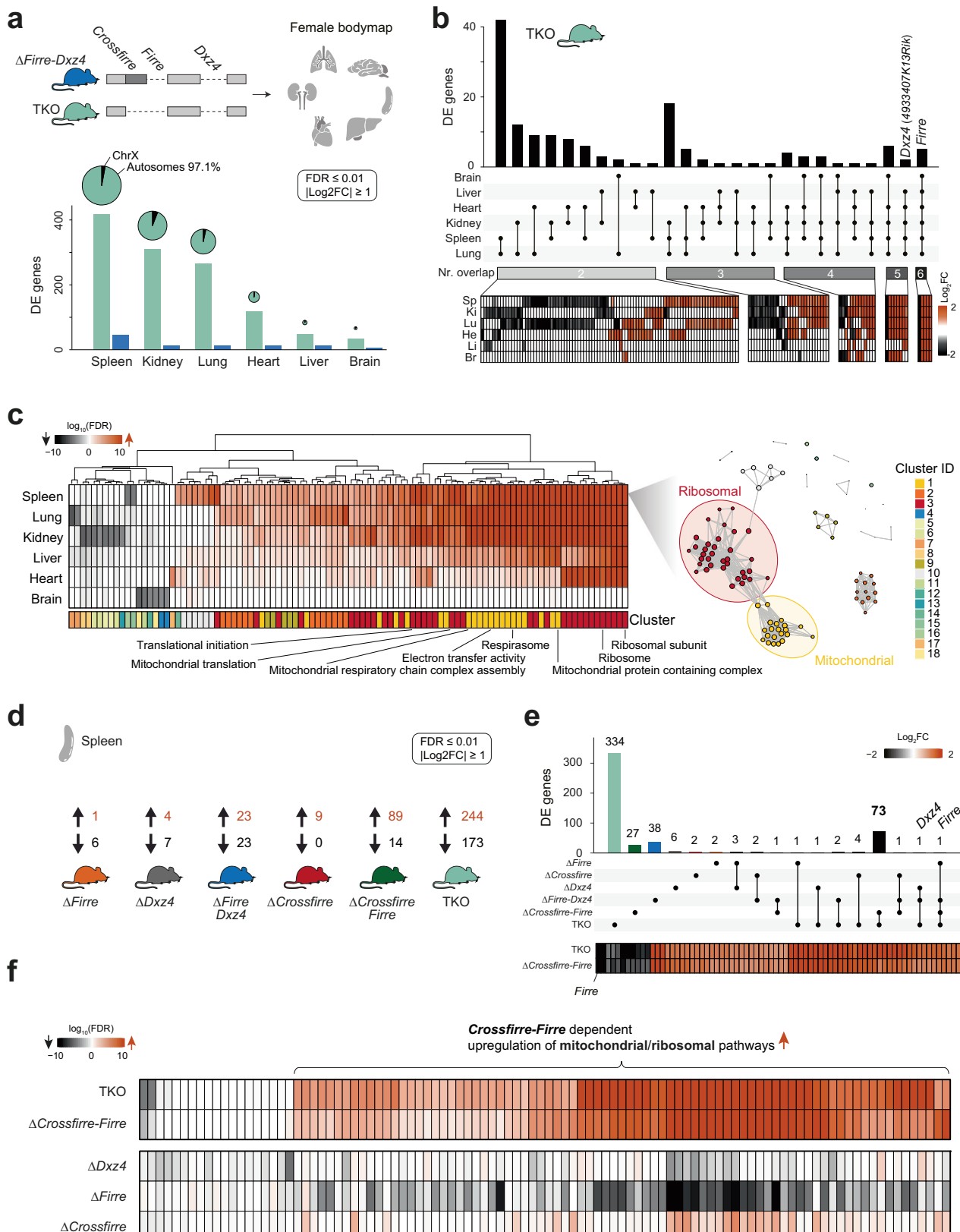

Immune cell analysis of the peripheral blood revealed apparent differences in the proportions of monocytes and B cells, as well as the CD4 / CD8 T cell ratio. Notably, our spleen scRNA-seq analysis also revealed differences in cell type composition, including B and CD4 T cell proportions (Supplementary Fig. 6a, b).

Besides the knockout-dependent phenotypes, sex-specific effects were observed. Males displayed a total of 13 different phenotypes covering six physiological categories including (i) elevated levels of the proinflammatory cytokine IL-6, as determined by immunology assessments (ii) impaired auditory brainstem response, measured by

**Fig. 5 | Homozygous double deletion of *Crossfirre-Firre*, results in upregulation of mitochondrial and ribosomal pathways. a** Transcriptomic bodymap for six different organs from homozygous adult female Δ*Firre-Dxz4* and Δ*Crossfirre-Firre-Dxz4* (TKO) mice (wildtype *n* = 4; Δ*Firre-Dxz4* *n* = 4; TKO *n* = 3). Chart bars show the number of significantly differentially expressed genes in the spleen, kidney, lung, heart, liver, and brain for TKO and Δ*Firre-Dxz4* (DEseq2: FDR ≤ 0.01, |log2FC| ≥ 1). Pie plots represent the proportion of differentially expressed genes between the X chromosome and autosomes as percentages. The size of the pie plots is proportional to the total amount of differentially expressed genes. **b** Number of differentially expressed genes shared across two, three, four, five, and six different tissues in TKO mice. The heatmap below shows the log_2fold changes of the overlapping dysregulated genes per tissue being up- (orange) or downregulated (black). **c** Heatmap showing the log_10(FDR) of the top 100 significantly enriched gene sets

from TKO gene set enrichment analysis (left; FDR ≤ 0.1). Network plot for spleen showing 18 gene set clusters with two dominant groups associated with mitochondrial (cluster ID: 1 *n* = 21) and ribosomal (cluster ID: 3 *n* = 35) gene sets (right). **d** Differential gene expression results from RNA-seq data obtained from female spleens of the different knockout strains. The number of significantly up- and downregulated genes is shown per genotype. **e** Number of significantly differentially expressed genes unique for each genotype and shared by the different knockout models. Below, heatmap showing log_2fold changes of differentially expressed genes shared between Δ*Crossfirre-Firre* and TKO mice. Color shading indicates up- (orange) and downregulation (black). **f** Gene set enrichment analysis of dysregulated genes from TKO, Δ*Crossfirre-Firre*, Δ*Dxz4*, Δ*Firre*, and Δ*Crossfirre* deletions. The heatmap shows the log_10(FDR) of all informative gene sets of the top 100 significantly enriched gene sets detected in (**c**).

neurological tests, and (iii) higher plasma insulin and triglyceride levels, coupled with decreased lactate and creatinine concentrations, as determined by clinical chemistry. Furthermore, dysmorphology screening (iv) revealed increased bone mineral content and 2/13 male mice showed abnormal digit hind paws. Metabolic tests (v) identified increased body weight, accompanied by higher oxygen consumption and metabolic rate, primarily due to the increase in lean mass. Additionally, a subset of male mutants (1/5) showed bronchopneumonia and mild inflammatory cell infiltrates in the pathology screen (vi) (Fig. 6b, c and Supplementary Data 1, sheet p).

In contrast, female mice showed higher eosinophil proportions, increased platelet volumes, and lower urea levels. Subtle effects were observed in a cardiology screening, including increased heart rates and heart-rate-corrected QT intervals. Moreover, 1/5 females showed pronounced dilated cardiomyopathy. Further histopathological examination revealed congestive arteries with thickening of the vascular wall and inflammatory cell surrounding in the lung. While these changes were apparent in 1/5 female mice, the effects were mild and focal in the remaining females (Fig. 6b, c and Supplementary Data 1, sheet q). A complete set of raw measurements from the relevant phenotyping tests is available on the Phenomap website of the GMC (https://www.mouseclinic.de). Overall, the screening analysis revealed a broad range of phenotypic changes, highlighting the complex influence of these loci on various physiological processes. While 32.14% (*n* = 9) of the phenotypes were specific to the knockout condition, the majority displayed sex-specific characteristics (male TKO-specific: 46.43%, *n* = 13; female TKO-specific: 21.43%, *n* = 6). This extensive phenotypic investigation of mutants lacking three X-linked lncRNAs, provides comprehensive insights into the putative role of the topmost sex-specific loci and lays a solid foundation for further phenotypic and mechanistic characterization.

## Discussion

In this study, we investigated the in vivo roles of *Crossfirre*, *Firre*, and *Dxz4* at the molecular and phenotypic level using multi-omic approaches in mouse models lacking these loci individually and in combinations. Our study confirmed the previously observed female-specific chromatin accessibility of *Crossfirre*, *Firre*, and *Dxz4*, known to originate from Xi[9,11,15,16]. Remarkably, we identified this pattern as the most female-specific epigenetic signature genome-wide, a finding that was consistent across organs. Furthermore, *Crossfirre* was previously identified as a maternally expressed lncRNA on the X chromosome, suggesting its potential involvement in imprinted XCI. Yet, despite the presence of female-specific signatures and imprinting of *Crossfirre*, we found that the absence of these loci did not affect imprinted XCI in the placenta, regardless of whether the deletions were on the maternally (Xa) or paternally (Xi) inherited X chromosome. Our allele-specific single-cell analysis allowed us to study how the absence of these Xi-specific epigenetic features and Xa-specific lncRNA expression affects random XCI maintenance in vivo. Consistent with the results for imprinted XCI, we did not observe differences in the allelic ratios of

X-linked genes. Our study further sheds light on the significance of LINE clusters in maintaining XCI, as the *Crossfirre* deletion includes an X-linked LINE cluster, elements that have been proposed to prevent gene escape[21,22]. However, no alterations in the allelic ratios were observed in proximity to the LINE cluster deletion, suggesting no impact on the escape of neighboring genes. In summary, we conclude that *Crossfirre*, *Firre*, and *Dxz4* deletions do not significantly affect random and imprinted XCI initiation and maintenance in vivo. However, we cannot rule out potential effects beyond the adult stage we investigated, particularly during aging and disease.

Despite the unique characteristics of these loci, we discovered that the absence of *Crossfirre*, as well as in combination with *Firre* and *Dxz4*, is not essential for mouse development. However, the extensive phenotypic characterization of TKO mutants revealed diverse phenotypes with the majority observed in a sex-specific manner. For example, males exhibited behavioral and neurological phenotypes, including impaired hearing, a characteristic of sensory processing disorders[30]. This finding is consistent with the intellectual disability phenotype reported in two independent case studies of male patients with duplications of the *Firre* locus[31,32]. Interestingly, one of the referenced studies also reported cardiac abnormalities[31], which may be in line with the subtle female-specific cardiovascular phenotypes observed in our phenotypic characterization. Analysis of immune cells in the peripheral blood confirmed the previously reported role of *Firre* in hematopoiesis and immune response[19,33,34]. Considering that these phenotypes were observed only in males, we can exclude a role for the Xi-specific epigenetic signatures, suggesting an RNA-mediated mechanism, as previously demonstrated for *Firre*[19]. In contrast, female-specific phenotypes were also observed, including higher eosinophilic proportions and increased platelet volumes, along with lower urea levels. These differences may suggest a role for the female-specific epigenetic signatures present on Xi. Evidence that the lack of these loci on Xa or Xi can result in different phenotypic outcomes was obtained from our scRNA-seq analysis of the spleen. We find that cells carrying the deletion on Xa, and thus lacking lncRNA expression, show a significant reduction in the proportion of B cells, consistent with the phenotypic immunology screen. In contrast, when the deletions were present on Xi, CD4 T cells were significantly reduced while B cells remained unaffected. These results highlight different functional characteristics of these loci on Xa and Xi that may explain sex-specific phenotypes. While the ability of Xi to influence Xa and autosomes has been recently reported[35–37], additional investigations are warranted to elucidate the underlying mechanisms.

Our transcriptomic analysis in the TKO revealed upregulation of mitochondrial and ribosomal gene sets, suggesting a role in energy metabolism[38,39]. Consistent with these expectations, we observed several phenotypes that may be related to these molecular findings. Among these are decreased urea levels and plasma cholesterol concentrations, which may be attributed to altered protein metabolism[40]. We also found decreased creatine levels and lactate concentrations with increased triglyceride levels, further supporting shifts in energy

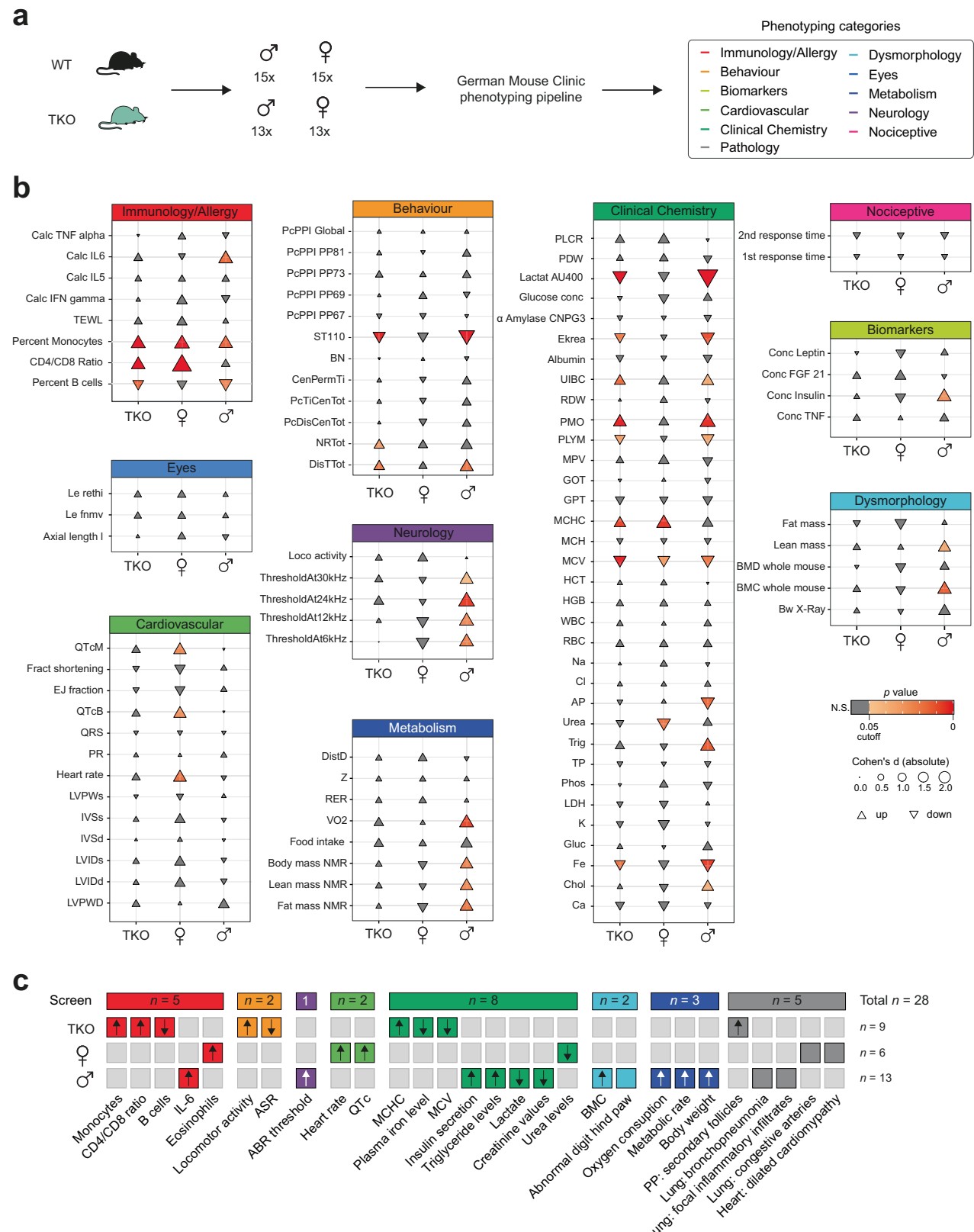

**Fig. 6 | Large-scale phenotyping analysis of TKO mutants uncovers knockout and sex-specific phenotypes. a** The German Mouse Clinic (GMC) phenotyping pipeline was conducted using 30 wildtype (WT, 15 males, 15 females) and 26 Δ*Crossfirre-Firre-Dxz4* (TKO) mice (13 males, 13 females). The pipeline covered the screening tests from the following categories: immunology/allergy, behavior, biomarkers, cardiovascular, clinical chemistry, pathology, dysmorphology, eyes, metabolism, neurology, and nociception. **b** Visualization of key measured parameters to provide a general overview of the GMC screening pipeline results. The

size of each triangle corresponds to the absolute effect size, represented by Cohen's d. Triangles pointing up or down indicate upregulation or downregulation, respectively. Parameters with a *p* value < 0.05 are considered significant. N.S.: not significant (*t*-test). Additional parameter information, including *p* values, effect sizes (Cohen's d, Hedges' g), and parameter abbreviations are provided in Supplementary Data 1, sheets o–q. **c** Concise overview of the significant parameters as phenotypes for the groups TKO (*n* = 9), female-specific (*n* = 6), and male-specific (*n* = 13) by phenotyping category.

metabolism[41–43]. Notably, the reduction in plasma lactate levels was one of the strongest measurements made by the GMC out of nearly one hundred knockout screens, including primarily protein-coding genes. The high degree of shared dysregulated autosomal genes between Δ*Crossfirre-Firre* and TKO suggests that the *Crossfirre-Firre* double deletion may be the primary contributor to the observed phenotypes. In order to regulate autosomal genes, an X-linked lncRNA would be required to function in *trans*. This is consistent with the *trans*-acting role of the *Firre* RNA, proposed to convey its function through *trans*-chromosomal associations[18,19]. Therefore, the more pronounced autosomal effect observed in the *Crossfirre-Firre* double deletion could be due to an RNA-mediated mechanism of *Crossfirre* itself or indirectly through the regulation of *Firre* via RNA- and DNA-mediated mechanisms or through the act of transcription. However, we did not observe a correlation between the expression levels of *Crossfirre* and *Firre* across organs, which may exclude a co-regulatory role in the investigated organs. Nevertheless, the mechanism underlying the observed autosomal dysregulation, and whether it is direct or indirect, remains to be elucidated.

In conclusion, our study provides an in-depth characterization of highly complex lncRNA loci that serve as a platform for the largest chromatin structures in mammals with prominent sex- and allele-specific epigenetic signatures. Despite the unique characteristics observed on the X chromosome at multiple layers, our findings reveal an interplay of *Crossfirre* and *Firre* in regulating autosomal genes, rather than affecting X inactivation biology. Our extensive molecular and phenotypic study provides a solid foundation for further exploring the intricate interplay of these conserved X-linked loci in vivo.

## Methods

### Mouse strains

All animal experiments were approved by Harvard University IACUC protocol (28–21) and were performed according to relevant guidelines, including the EU guideline 2010/63, ARRIVE guidelines, and the German Animal Welfare Act (Kreisverwaltungsreferat der Stadt München, Veterinäramt München Stadt). C57BL/6J (BL6), B6D2F1/J (F1 BL6 and DBA), and CAST mice were purchased from the Jackson Laboratory, and CD-1 females from Charles River. Mice were maintained in pathogen-free conditions at Harvard University's Biological Research Infrastructure and Institute of Pharmacology and Toxicology at Technical University Munich with a 12-h light cycle, a temperature of $21 \pm 1$ °C and 40−70% humidity. Generation of *Crossfirre* and *Crossfirre-Firre* deletions were performed as previously described for *Dxz4* and *Firre*[14]. Briefly, superovulated B6D2F1 females were mated with BL6 males, and pronuclear stage 3 (PN3) zygotes were isolated, followed by co-injection of Cas9 mRNA (200 ng/μl) combined with two gRNAs spanning each target locus (50 ng/μl, Supplementary Fig. 2a; Supplementary Data 1, sheet d). The zygotes were cultivated to the blastocyst stage and transferred into pseudopregnant CD-1 females[14,44]. Offspring carrying the deletion were identified by PCR using primers spanning the deleted regions and subsequent Sanger sequencing of the PCR product (Supplementary Fig. 2a–c; Supplementary Data 1, sheet d). To obtain TKO mice, Δ*Dxz4* females (ref. 14, originated from clone 8) were mated with Δ*Crossfirre-Firre* males (originated from clone 42) to generate females carrying both deletions on separate X chromosomes. Next, we crossed these females to BL6 males and screened for meiotic recombination events between *Dxz4* and *Crossfirre-Firre* to have both deletions in phase on the same X chromosome. For the transcriptomic analysis, all founder mice (with 75% BL6 background) underwent at least two backcrosses with BL6 to eliminate CRISPR-Cas9 off-target effects and further minimize strain background influence (expected 93% BL6 background). To ensure consistent strain background, WT control

mice were obtained by backcrossing the founder mice. For the large-scale phenotyping analysis of TKO mutants, we conducted two additional backcrosses of the TKO mutants, resulting in a total of five backcrosses (expected 98% BL6 background).

### Tissue isolation and library preparation for bulk and scRNA-seq

To determine the effect of the investigated loci on imprinted XCI, placentas were isolated at E12.5 from F1 reciprocal crosses between the mutant strains (Δ*Crossfirre*, Δ*Crossfirre-Firre*, and TKO) and CAST. Three biological replicates were isolated from placentas carrying the deletion on the maternal Xa or paternal Xi. WT control placentas as well as Δ*Dxz4*, Δ*Firre*, and Δ*Firre-Dxz4* were reanalyzed from ref. 14. In total, this results in the following number of biological replicates per genotype: Δ*Crossfirre*: Xa $n = 3$, Xi $n = 3$; Δ*Firre*: Xa $n = 3$, Xi $n = 3$; Δ*Dxz4*: Xa $n = 3$, Xi $n = 3$; Δ*Firre-Dxz4*: Xa $n = 3$, Xi $n = 3$; Δ*Crossfirre-Firre*: Xa $n = 3$, Xi $n = 3$; Δ*Crossfirre-Firre-Dxz4*: Xa $n = 3$, Xi $n = 3$; WT: BL6xCAST $n = 8$, CASTxBL6 $n = 9$.

To generate a transcriptomic bodymap, adult organs (spleen, kidney, lung, heart, liver, and brain) were isolated from 6-week-old TKO mice ($n = 3$). Additionally, the spleen was harvested from Δ*Crossfirre* ($n = 3$) and Δ*Crossfirre-Firre* ($n = 2$) mice mutants. Δ*Dxz4*, Δ*Firre*, and Δ*Firre-Dxz4*, as well as WT control adult organ samples, were reanalyzed from ref. 14. The collected tissues were snap-frozen and stored at −80 °C until further use. The RNA was extracted from TRIzol using RNeasy mini columns (Qiagen, #74104). PolyA+ mRNA libraries were generated from total RNA using the Illumina TruSeq kit. Strand-specific libraries were created for F1 placentas, and non-stranded libraries for adult organs. Library concentrations were quantified on a Qubit 2.0 Fluorometer. Purity and fragment size were assessed on an Agilent 2100 Bioanalyzer. Libraries were sequenced on a HiSeq 2500 at Harvard University's Bauer Sequencing Core (75 bp paired-end).

To test whether the absence of *Crossfirre*, *Firre*, and *Dxz4* has an impact on random XCI, spleens from heterozygous TKO (TKO −/+ × CAST) females and WT littermates (TKO +/+ × CAST) were isolated from 6-week-old F1 followed by single-cell RNA-seq. To obtain a single-cell suspension, spleens were isolated on ice and homogenized between two glass slides. The single-cell suspension was filtered through 70 μm and 30 μm strainers and incubated with Fc-block for 15 min. Then cells were stained with Zombie Green (Viability, BioLegend, #423111) and a TER-119-PE antibody (Erythrocytes, Thermo-Fisher, #12-5921-83). After staining, cells were incubated with Cell Multiplexing Oligos (10x, #PN-1000261) to add a specific barcode to each sample, allowing the pooling of all samples in one 10x reaction. The Zombie Green-negative and TER-119-negative population was FACS-sorted (Supplementary Fig. 7a, b) and counted for subsequent single-cell library generation using the Chromium Next GEM Single Cell 3′ Reagent Kits v3.1 (Dual Index, 10x, #PN-1000269) with Feature Barcode technology for Cell Multiplexing. To verify the XCI skewing ratio observed in single-cell RNA sequencing, spleens from 6-week-old TKO heterozygous females ($n = 3$) and WT littermates ($n = 3$) were isolated, snap-frozen and stored at −80 °C. RNA was extracted using Qiagen's TRIzol reagent and bulk RNA-sequencing libraries were generated using Illumina's Stranded mRNA Prep, Ligation kit. Library concentrations were quantified on Agilent's TapeStation System. Both the scRNA-seq as well as the bulk for the F1 spleens were sequenced on a NovaSeq6000 at Helmholtz Munich (50 bp paired-end).

### Bulk RNA-seq preprocessing and analysis

RNA-seq data was inspected by the FASTQC software tool and mapped to the GENCODE_M25GRCm38.p6_201911[45] primary assembly using the STAR aligner (v.2.6.0c[46]). Reads with an intron size > 100,000, multi-mappers, and alignments that contain non-canonical junctions were excluded for downstream analysis. The remaining uniquely aligned reads were quantified using htseq-count (HTSeq v.0.11.3[47]). We used

the GENCODE vM25 primary assembly[45] as an annotation file and manually added *Crossfirre* (*Gm35612*) from the RefSeq gene annotation[48]. Due to the unstranded nature of adult organ libraries, the last exon of the *Crossfirre* locus, overlapping *Firre* in the antisense direction was removed for read counting. Strand-specific brain RNA-seq data from ref. 15 was further separated according to strand using a custom Perl script (SRR3085966, SRR3085967, SRR3085968, SRR3085969). SNPsplit v0.3.2 was used to further assign the strand-specific reads to the corresponding allele[49], as well as to process the publicly available H3K4me3 sequencing data from female mouse embryonic fibroblasts (SRR2038034 SRR2038035, SRR2038036, SRR2038037)[50].

We calculated transcripts per million (TPMs) using custom R scripts to normalize the gene expression for sequencing depth and gene length. Differential gene expression analysis was conducted using DESeq2 (version 1.32.0[51]). Dysregulated genes were called significant with an FDR ≤ 0.01 and an adjusted |log2FC| ≥ 1 (lfcShrink apeglm[52]).

## GSEA
GSEA was performed based on the DESeq2 test statistics and cameraPR() of the limma R package[53]. Gene ontology sets were derived from the MSigDB (c5.go.v7.4.symbols[54]) and called significant with an FDR-adjusted $p$ value ≤ 0.1. Gene sets with ≤10 or ≥500 genes were removed for downstream analyses. The top 100 gene sets were extracted based on the smallest FDR values of all gene sets.

Network plots were generated for the top 100 dysregulated gene sets using the simplifyEnrichment[55] and igraph R package[56] to calculate a similarity matrix based on genes. Clustering the gene sets was performed by simplifyEnrichment's cluster_terms() and the walktrap method[55].

## ATAC-seq data processing and analysis
Public ATAC sequencing data from ref. 23 (SRR8119821, SRR8119822, SRR8119826, SRR8119827, SRR8119832, SRR8119833, SRR8119834, SRR8119835, SRR8119836, SRR8119837 SRR8119838, SRR8119839, SRR8119850 SRR8119851, SRR8119852, SRR8119853 SRR8119854, SRR8119855, SRR8119856, SRR8119857, SRR8119858, SRR8119859 SRR8119864, SRR8119865) and from ref. 24 (SRR3933589, SRR3933595) were aligned with the bowtie2 aligner in paired-end mode by default parameters[57]. The reference genome was built from the GENCODE_M25GRCm38.p6_201911[45] reference using bowtie2-build. Quality control of mapped reads included: Removal of mitochondrial reads and mapping artifacts with base-pair length ≤38 or ≥2000, filtering of low-quality reads with MAPQ < 20 and ENCODE blacklist genes (blacklist.v2.bed[58]), and exclusion of duplicates identified by GATK MarkDuplicates (v 4.1.0.0[59]). Broad peaks were called from the quality-controlled data using MACS2 callpeak[60].

To assess differences in the chromatin profile of male and female samples, we intersected broad peaks called by MACS2 for each organ and sex. We then counted the number of peaks within sliding windows of 100 Kb across the entire genome. The intervals between the windows were 50 Kb. The median number of peaks per window was calculated across organs and sex and used to calculate $\log_{10} p$ values from a binomial test to assess significant differences. A positive value was assigned if more peaks were present in females than in males, while negative values were assigned if the reverse was true.

## Allele-specific analysis for RNA- and ATAC-seq data
Allele-specific analysis for RNA-seq and ATAC-seq data was performed using the Allelome.PRO pipeline[50].

For the allele-specific RNA-seq analysis, we used the GRCm38/mm10 RefSeq gene annotation[48] (downloaded in February 2018) in conjunction with the previously generated CAST/BL6 SNP annotation ($n = 15,438,314$)[14,61]. This SNP file contains only CAST/BL6 SNPs in which the BL6 allele is shared with DBA, BALB/C and 129 strains, to minimize confounding effects from strain backgrounds[14]. SNPs were included if the coverage was ≥1 read, and genes were removed if the total read count was <30 for RNA-seq data.

For the allele-specific ATAC-seq analysis of clonal F1 neural progenitor cells[24], we used a 50 Kb sliding window annotation of the mm10 genome and 20,563,466 SNPs between 129S1/SvImJ and CAST obtained from the Sanger database[61]. A minimum SNP coverage of ≥1 read was applied, along with a total read count of ≥50 reads per genomic window.

## scRNA-seq preprocessing
Demultiplexing, alignment, and read count quantification of the raw sequencing data were performed using the *cellranger multi* pipeline (Cell Ranger 6.1.2 toolkit, 10x Genomics[62]). As reference data, we implemented the 10x Genomics pre-built Cell Ranger reference package for the mm10 genome (version 2020-A).

Further processing of individual sample matrices was done using R and the Seurat 4.1.1 package[63]. Quality control was performed on merged samples by filtering cells with nFeature_RNA > 500 and nFeature_RNA < 5000, nCount_RNA > 2000, and nCount_RNA < 20,000, and percent of mitochondrial reads <10. Additionally, genes were filtered to be expressed in >10 cells. Seurat objects were further normalized using SCTransform while regressing for mitochondrial read percentage. The heterozygous TKO and WT data were integrated using Seurat and the IntegrateData() function. The top 40 principal components were calculated using RunPCA() for UMAP dimensionality reduction and clustering. Individual cell clusters were annotated manually based on gene marker expression.

## Allele-specific scRNA-seq analysis
We performed an allele-specific analysis to investigate whether the CAST or BL6 allele of the X chromosome is active. Single-cell bam files were generated with filterbarcodes from the sinto package (https://timoast.github.io/sinto/index.html). Allelome.PRO was used to calculate allelic ratios for each chromosome, using the same SNP file as for the bulk RNA-seq analysis and a custom-made chromosomal annotation file. Single-cell transcriptomes were removed for downstream analysis if the total read number over the X chromosome was <10 ($n = 3685$). Based on the allelic ratios, cells were classified according to the status of the X chromosome (allelic ratio ≥ 0.7 CAST Xa; allelic ratio 0.3 ≤ BL6 Xa).

In order to perform an allele-specific pseudobulk analysis for the scRNA-seq data, we summarized cells from TKO and WT with either the CAST or BL6 allele active. Subsequently, we used Allelome.PRO to perform the allele-specific analysis using the identical SNP and annotation file as described for the bulk RNA-seq data. The minimum number of reads overlapping SNPs to be included was 1, while the total read cutoff was ≥30.

## Phenotypic analysis of the German Mouse Clinic
We subjected mutants along with their corresponding WT controls (females: 13x TKO, 15x WT, males: 13x TKO, 15x WT) to the GMC to perform a primary phenotypic analysis[29,64]. Animals were maintained in IVC cages with water and standard mouse chow according to the directive 2010/63/EU, German national laws, and GMC housing conditions. All tests performed were approved by the responsible authority of the district government of Upper Bavaria. Mice were confirmed to be pathogen-free according to the FELASA recommendations. If not stated otherwise, data that was generated by the GMC was analyzed using R (version 3.6.3). Genotype effects were tested by performing $t$-test, Wilcoxon rank sum test, linear models, or ANOVA and posthoc tests, or Fisher's exact test, based on the assumed parameter distribution. Phenotypes were declared significant with a $p$ value < 0.05, which was not corrected for multiple testing.

**Reporting summary**

Further information on research design is available in the Nature Portfolio Reporting Summary linked to this article.

## Data availability

The sequence data generated in this study have been deposited in the Gene Expression Omnibus (GEO) database under accession code GSE219160. Raw measurements from the relevant phenotyping tests are available on the Phenomap website of the GMC (https://www.mouseclinic.de). Source data are provided with this paper.

## Code availability

The relevant code to reproduce the analyses and figures of this study is available at https://github.com/AndergassenLab/DissectX.

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

## Acknowledgements

Sequencing was performed at the Bauer Core Facility at Harvard University and the Genomics Core Facility at Helmholtz Munich. The authors thank Anne Dueck for assistance with the 10x genomics system. We would like to thank the GMC technician team and the Core Facility Laboratory Animal Services at Helmholtz Munich for their expert support. This work was partly supported by the Deutsche Forschungsgemeinschaft (Project ID: 403584255—TRR 267, D.A. and S.E.), the NIH (P01 GM099117, J.L.R and A.M.), the BMBF in the framework of the Cluster4Future program CNATM (Project ID: 03ZU1201BA, D.A. and S.E.), the DZHK (Junior Research Group, D.A.), the German Federal Ministry of Education and Research (Infrafrontier grant 01KX1012, M.H.d.A.) and the German Center for Diabetes Research (DZD, M.H.d.A.).

## Author contributions

Tim P. Hasenbein: conceptualization, data curation, software, formal analysis, investigation, visualization, methodology, writing—original draft, writing—review and editing. Sarah Hoelzl: conceptualization, investigation, resources, methodology, writing—original draft, writing—review and editing. Zachary D. Smith: resources, investigation, methodology, writing—review and editing. Chiara Gerhardinger, Marion O. C. Gonner: investigation, writing—review and editing. Antonio Aguilar-Pimentel, Oana V. Amarie, Lore Becker, Julia Calzada-Wack, Nathalia, R. V. Dragano, Patricia da Silva-Buttkus, Lillian Garrett, Sabine M. Hölter, Markus, Kraiger, Manuela A. Östereicher, Birgit Rathkolb, Adrián Sanz-Moreno, Nadine Spielmann: mouse phenotyping and data analysis and interpretation. Wolfgang Wurst: supervising of mouse phenotyping project. Valerie Gailus-Durner, Helmut Fuchs: supervising of mouse phenotyping project, conceptualization of mouse phenotyping tests. Martin Hrabě de Angelis: conceptualization of mouse phenotyping tests. Alexander Meissner, Stefan Engelhardt: writing—review and editing. John L. Rinn: conceptualization, supervision, funding acquisition, writing—review and editing. Daniel Andergassen: conceptualization, supervision, funding acquisition, project administration, data curation, investigation, formal analysis, validation, methodology, writing—original draft, writing—review and editing.

## Funding

## Competing interests

The authors declare no competing interests.

## Additional information

[1]Institute of Pharmacology and Toxicology, Technische Universität München, Munich, Germany. [2]DZHK (German Centre for Cardiovascular Research), Partner Site Munich Heart Alliance, Munich, Germany. [3]Department of Stem Cell and Regenerative Biology, Harvard University, Cambridge, MA, USA. [4]Broad Institute of MIT and Harvard, Cambridge, MA, USA. [5]Yale Stem Cell Center, Department of Genetics, Yale School of Medicine, New Haven, CT, USA. [6]Institute of Experimental Genetics, German Mouse Clinic, Helmholtz Zentrum München, Neuherberg, Germany. [7]Institute of Developmental Genetics, Helmholtz Zentrum München, Neuherberg, Germany. [8]TUM School of Life Sciences, Technische Universität München, Freising, Germany. [9]Institute of Experimental Genetics, Applied Computational Biology, Helmholtz Zentrum München, Neuherberg, Germany. [10]German Center for Diabetes Research (DZD), Neuherberg, Germany. [11]Institute of Molecular Animal Breeding and Biotechnology, Gene Center, Ludwig-Maximilians-Universität München, Munich, Germany. [12]Department of Developmental Genetics, TUM School of Life Sciences, Technische Universität München, Freising, Germany. [13]Deutsches Institut für Neurodegenerative Erkrankungen (DZNE) Site Munich, Munich, Germany. [14]Department of Experimental Genetics, TUM School of Life Sciences, Technische Universität München, Freising, Germany. [15]Department of Genome Regulation, Max Planck Institute for Molecular Genetics, Berlin, Germany. [16]Institute of Chemistry and Biochemistry, Freie Universität Berlin, Berlin, Germany. [17]BioFrontiers Institute, University of Colorado Boulder, Boulder, CO, USA. [18]Department of Biochemistry, University of Colorado Boulder, Boulder, CO, USA. [19]These authors contributed equally: Tim P. Hasenbein, Sarah Hoelzl. ✉e-mail: John.Rinn@Colorado.EDU; daniel.andergassen@tum.de

