## [Peer Review file · Nature Communications]

X-linked deletion of Crossfirre, Firre, and Dxz4 in vivo uncovers diverse phenotypes and combinatorial effects on autosomes

Corresponding Author: Dr Daniel Andergassen

Version 0:

Reviewer comments:

Reviewer #1

(Remarks to the Author)

In this manuscript the authors Hasenbein*, Hoelzl* et al. explore the roles of the X-linked lncRNAs Firre, Crossfire, and Dxz4 through separate and combined deletions in mice. The authors initially hypothesize the involvement of these loci in X-inactivation in both the embryo proper and extraembryonic tissues, but subsequently refute these hypotheses. Instead, they report broad and pleiotropic phenotypes associated with autosomal effects, which manifest both independently of sex (mutants vs. WT) and in a sex-dependent manner (males vs. females) in vivo. They conduct mouse genetic deletions and validations to a good standard, and provide detailed and extensive characterisation of resulting phenotypes across different organs and via diverse methods. The reviewer appreciates the financial and experimental effort invested in the manuscript. However, there are several major issues recurring throughout the manuscript:

1. While a lot of work has gone into the manuscript, I'm not sure what the take away message is regarding the functions of these non-coding RNAs. The reported effects are pleiotropic, but no molecular mechanism is offered. I do understand though that this is the first manuscript to delete all three ncRNAs and as such carries the burden of describing phenotypes at a broad level. Maybe the authors should be allowed to advocate on this issue
2. There is a seeming lack of understanding of the literature or overlooked results of major importance in the field. This is especially problematic given that it affects the first sentence in the manuscript abstract, namely: "The lncRNA Crossfirre was identified as the only imprinted X-gene" This statement is incorrect. Xist, and other genes in the X-inactivation centre, Ftx and Jpx, are notable examples, as are RhoX5 (2006, PMID: 16431368) and Fthl17 (PMID: 20185572). It might have been that the authors were referring specifically to imprinted X-genes expressed on the maternally inherited X chromosome, but even in that case, some genes have already been reported, e.g., in 2005 Xlr3b (PMID: 15908950), Xlr4b and 4c (PMID: 15908953)
3. Overall, the flow of the manuscript lacks clarity and coherence. The reviewer found it challenging to follow the logical trail of the text and observed a disjointed progression from one experiment to another. Lacking citations and correspondence with previous accounts in the literature has also contributed to this (see point 1). The figures are beautifully presented, but are often not clear to understand and interpret.

Minor points:

Unclear sentences:

L86: "Has been identified in a recent annotation" Please provide a reference.

L92: Why would a gene expressed specifically on the maternal X warrants further investigation for a link to XCI? For the reviewer this is not obvious. Please explain.

L100-104: why were the authors expecting more of an X specific phenotype?

L215: "We detected a more pronounced skewing ration in the TKO heterozygous spleen cell population" The reviewer wants to know if this finding is significant. If so, it should be interesting to discuss and investigate.

L371: "This study found that Firre RNA expressed from the Xa chromosome maintains histone H3K27me3 enrichment on the Xi without reactivating X linked genes". For the reviewer is this an unclear sentence.

2. Unclear figures and analyses:

Figure 1b and c. The axis legend : $\text{Log}_{10}(p)$ is wrong, in both b and c as well as in the material and method, as the authors clearly plotted “ $1-\log_{10}(p)$ ”. The analysis, particularly regarding how the sliding window is defined, needs to be more thoroughly explained. Furthermore, previous studies (ref.11 in the paper) have already demonstrated that Firre is the only domain producing accessibility peaks on the Xi in ATAC-seq, and this finding should be addressed in the discussion.

Figure 2.d 50 pups from 8 litters are 6.25 pups per litter, 55 are 6.875, is there a reason why 6.25 is rounded to 6.3 but 6.875 is rounded to 6.8. This seems incoherent, please explain this.

Figure 3d. The reviewer finds it concerning that both controls (Xi and Xa deletion, but without deletion) show different allelic ratios violin plots. This casts doubts on the reproducibility of the authors allelic ratio determination. This should at least be discussed.

Figure 3e. It would be helpful if the authors could provide a list of genes they are plotting to facilitate further analysis of the graph. Additionally, the absence of Firre/Crossfire from the BL6 strain in the Xa-specific plot is not discussed. If Crossfire is maternally imprinted, its absence should result in no allelic bias in the plots when the deletion is on the Xa. This also applies for the Dxz4 KO.

Figure 4 analysis. The numbers do not add up: WT = 2046 but $1342+640+61=2043$ Heterozygous = 1648 but $1359+243+40=1642$.

Figure 4e. There seems to be a similar issue as in Figure 3. If DXz4/Firre and Crossfire are deleted from the active (Xa) BL6 chromosome, why are all reads for these genes depicted in black? If the triple knockout still shows expression from all genes on the active X chromosome, it raises concerns.

Figure 6. Difficult to interpret. The reviewer is unsure about the significance of a 1 Cohen's d unit and what the absence of an effect indicates.

In correspondence to the phenotype analysis in figure 6. It would be beneficial to explore any interesting phenotypes further, especially if they are reproducible and significant. It could be insightful to conduct more analyses beyond the hematopoietic cell lineage, especially considering this has already been addressed in a previous publication from the lab. In addition, investigating how these findings correlate with single-cell analysis of organs could provide interesting insights.

Reviewer #2

(Remarks to the Author)

Hasenbein et al. present a comprehensive analysis of three intriguing X-linked lncRNA loci, two of which produce convergently transcribed RNAs, Crossfire and Firre, and a third which is also transcribed into a lncRNA and like the Firre locus harbors a large number of embedded DNA regulatory elements. The major findings of the study are significant to both the lncRNA and X-chromosome inactivation fields. Despite the distinct epigenetic features associated with the Crossfire, Firre, and DX4 loci, and the presence of these loci on the X chromosome, their combinatorial deletion results in no obvious defect in XCI. Intriguingly however, tissue-specific phenotypes are observed on autosomal genes. The study is clearly presented and very well controlled. I found the methodology to be sound, that the conclusions were supported by the data, and that the methods were sufficiently detailed so that others could reproduce the analyses.

I noted that the authors referenced Supplemental Table 1 throughout the manuscript but I was unable to access this as a reviewer. My main request would be that the authors include in that table or another one like it the lists of differentially expressed genes, and their associated expression values in the tissues analyzed, if those data have not already been included.

Beyond a desire for well annotated lists of differentially expressed genes (which may already exist in Table S1), I have very little to suggest. If the authors felt it were appropriate to do so, in the discussion section they could speculate as to whether Crossfire and the other lncRNAs function as RNA products or if it is the act of their transcription that is more likely to be the result of the mutant phenotypes upon their deletion.

Reviewer #3

(Remarks to the Author)

The findings presented in this manuscript are very relevant for the non-coding genome research field as the authors underwent a detailed phenotyping at the molecular and macroscopic scales for non-coding transcribed loci and provide a robust pipeline to understand the functions of the non-coding genome. Moreover, the findings are relevant for the XCI field as the authors provide a clear demonstration of the absence of regulatory implication of Crossfire, Firre and Dxz4 in the regulation of the mouse XCI, a lasting question in the field. Publication of this study is recommended on the basis of these important demonstrations. We however have some doubts on the structure of the manuscript. As XCI aficionados, we were more compelled by the non-XCI related phenotypes and left hanging with a sense that links/analysis/discussions were missing between observed gene deregulations and cell/organ phenotypes. The comments hereafter further elaborates on this.

Comments:

1. The title “Dissecting the in vivo role of X-linked lncRNA loci with conserved sex- and allele-specific epigenetic

signatures" is too vague and does not reflect the results nor the conclusion of the manuscript. For instance, the conservation is not the topic of the manuscript moreover the sex and allele-specific epigenetic signatures of the loci under scrutiny are of very moderate interest regarding the authors findings. We feel that the title should name the loci under scrutiny, their trans acting role in autosomal gene regulation and their impact at the phenotypic level.

2. The authors often refer to Crossfire as the "only imprinted X-linked gene" or "Crossfirre, the only imprinted lncRNA located on the X chromosome". Care should be given in revising these statements; Xist, an X-linked gene, also has imprinted regulation, albeit paternally and not in the soma.

3. The authors write "we speculate that the locus [Crossfire] may serve as a marker to prevent silencing of the maternal X chromosome in extraembryonic lineages with imprinted XCI": maternal imprint of the Xist promoter has been fairly well established and as it is formulated, this hypothesis seems unlikely and does not need to be stated as is. Moreover, the authors and others previously shown that the Firre-Dxz4 loci is not regulating imprinted nor random XCI in mice. Altogether this XCI narrative undermines the very interesting findings of the role Crossfire-Firre-Dxz4 in autosomal transcriptional regulation and the thorough phenotyping study the authors have conducted. Hence, we would have favored a narrative centered around how X-linked loci can affect autosomal gene regulation and integrate their manuscript in a corpus of publications addressing those questions (Brenes 2021, Richart 2022, Roman 2024). We strongly feel the need to expand their analysis in the autosomal implication of the loci (see below).

4. Having the expression levels of Crossfire, Firre and Dxz4 across the different tissues in WT samples could help connect the strong phenotype they observe in the spleen and potentially speculate on the molecular mechanism. Indeed, the trans-acting effect suggest that the function is mediated through the RNA molecule produced from those loci.

5. The authors should use their scRNA-Seq spleen data to deepen the molecular exploration of the TKO phenotypes they undergone with bulk RNA-Seq, what cell populations are the most affected by the TKO ? How does it relate to the results of their phenotypic screen? Could it explain sex-specific effects?

6. The phenotyping of the mice TKO mutants is for us the most compelling aspect of the paper and should be further exploited and described both in the main text and in the discussion

1. In the main text: put p-values associated to the Cohen's d values to help the reader interpret the figures. Moreover, we suggest to the authors to add another measure of the effect size, the Hedge g, that is better powered for small sized cohorts. We believe the authors should try to connect the phenotype to the findings of their screen and the results of their multiple differential expression analysis. For example, how does TKO mice phenotypes relate to the enriched GO terms they are finding ? We strongly believe that the analysis of the scRNA-Seq spleen data could strengthen the conclusions related to the Immune cell analysis.

2. In the discussion: To our knowledge, the phenotyping approaches the authors have conducted is a first of its kind for non-coding elements and particularly lncRNAs. It would be nice to discuss in a couple of sentences how the results compare to similar studies working on protein-coding genes. The authors should discuss the general and sex-specific phenotypes they found in light of the sex-specific epigenetic patterns they describe in their first figures. Particularly, they are stronger effect size (Cohen's $d \geq 1$) for males than females, yet the epigenetic patterns they described are mostly female-biased.

Minor comments

Throughout the text TKO is employed but, in the figures, Δ TKO, which doesn't make sense, is used. Please remove the Δ from the Δ TKO captions.

SupF 1A: the RNA-seq track is not visually informative, a y-axis would help, or a simple count table. Y-axis would also be useful when looking at SupF 1B.

Fig2 C: Please put the wt or Δ under the axis. A floating Δ is not necessary.

Fig2D legend: Although we commend reproducible research, we do not feel inclusion of R formula for the determination of the p-value is appropriate.

Fig4A: Instead of RBC-, TER-119- would be more coherent with Zombie-

Reviewer #4

(Remarks to the Author)

Version 1:

Reviewer comments:

Reviewer #1

(Remarks to the Author)

The authors have done an excellent and thorough revision that have addressed all of our comments.

Reviewer #2

(Remarks to the Author)

the authors have addressed my prior requests and I would support publication of this study, which I believe describes results that are important for the XCI and lncRNA communities.

Reviewer #5

(Remarks to the Author)

The manuscript entitled "X-linked deletion of Crossfire, Firre, and Dxz4 in vivo uncovers diverse phenotypes and combinatorial effects on autosomal gene regulation" by Hasenbein et al. follows on the authors previous work on Firre and Dxz4 and presents a study of mice harbouring combinations of X chromosomal mutations in Firre, Dxz4, and Crossfire, which is an antisense transcript that partially overlaps Firre. All genotypes are viable and the triple mutation can be bred in homozygous state suggesting subtle effect on development. This is consistent with the authors earlier work on homozygous Firre and Dxz4 double mutations in mice but also extends by including Crossfire.

The authors have expectations that any of these genes could be regulating X inactivation in redundant manner but an analysis of X-linked allelic gene expression does not find evidence for this hypothesis. Notably, effects on autosomal gene expression is observed suggesting a role in gene regulation. The authors go on to perform an unbiased phenotyping using a service. Although, several phenotypes are recorded, the overall connection and conclusion remains a bit unclear and it could be made more clear what is biologically relevance as the mice seem overall healthy.

The manuscript stands out for the large amount of data. Although some constitutes negative evidence an advance is made over the authors earlier analysis that did not include Crossfire (both Super Loup and Megadomain formation were already abrogated by Firre/Dxz4 mutations in the authors previous work.) The relevance of the results can likely be increased by adding to the text to the effect that the view of Crossfire as a major regulator of imprinting and XCI might need correction.

Specific points

1. Fig 6. A focus of the relevant phenotyping data would make the study accessible to a wider readership. The current data presentation will not be useful for a wider readership and the expert would take advantage of seeing the histology analysis and other types of experimental data from the various assay). This would allow the authors also to include targeted analysis of certain aspects. In particular FACS analysis of the spleen populations. The present Fig 6 is certainly very useful way to summarize for the database and best be included as supplement along with the statistical view.

2. line 164: Fig 2d has a table from which it might appear that the litter size of the TKO is increased (TKO 9.4, delta Crossfire 6.3, delta Crossfire and Firre 6.9, wild type missing). In the context that TKO display subtle phenotypes, the increased litter size might be noteworthy. Is this statistically significant? If the TKO mice are fitter, is there any indication that the DXZ4 / Firre system might be a parasitic genetic system?

3. The effect of the mutations on allelic expression suggests that imprinted and random XCI are comparable to controls. The authors use this to argue against a function in XCI. However, subtle effects on stability of chromatin or repression on the Xi might be missed. From Ciz1 mutant mice and conditional Xist mutation in mice would be blood cell hyperplasia (splenomegaly) or tumours at old age. It would be important to include a statement if aged mice were analysed and such blood cell issues ruled out. This would strengthen the idea of dispensability for maintenance of Xi repression.

Minor points

a) Crossfire is described by the authors as an imprinted gene which often affect embryonic growth in a subtle manner. It would be interesting to investigate if there were a growth effect during transient embryonic growth period, which often can be compensated as development progresses towards birth.

b) The authors show that different autosomal genes are shown to be misregulated by (cross) firre and DXZ4. Conversely, has an analysis been performed to of the ncRNAs affect autosomal genes via miR regulation (sponge)? Or are these misregulated genes related to transposable or mobile genetic elements?

Version 2:

Reviewer comments:

Reviewer #5

(Remarks to the Author)

The authors have have addressed all of my comments in their comprehensive response. In particular, the availability of the large amount of phenotyping data is an important asset of the study, which will be of high interest to researchers in noncoding RNA, gene regulation and mouse development.

We thank the reviewers for their time and diligence in reviewing our manuscript and greatly appreciate their valuable comments. We have provided point-by-point responses to the comments below and are confident that the revisions have significantly improved the manuscript.

Reviewer #1

In this manuscript the authors Hasenbein*, Hoelzl* et al. explore the roles of the X-linked lncRNAs *Firre*, *Crossfire*, and *Dxz4* through separate and combined deletions in mice. The authors initially hypothesize the involvement of these loci in X-inactivation in both the embryo proper and extraembryonic tissues, but subsequently refute these hypotheses. Instead, they report broad and pleiotropic phenotypes associated with autosomal effects, which manifest both independently of sex (mutants vs. WT) and in a sex-dependent manner (males vs. females) *in vivo*. They conduct mouse genetic deletions and validations to a good standard, and provide detailed and extensive characterisation of resulting phenotypes across different organs and via diverse methods. The reviewer appreciates the financial and experimental effort invested in the manuscript. However, there are several major issues recurring throughout the manuscript:

1. While a lot of work has gone into the manuscript, I'm not sure what the take away message is regarding the functions of these non-coding RNAs. The reported effects are pleiotropic, but no molecular mechanism is offered. I do understand though that this is the first manuscript to delete all three ncRNAs and as such carries the burden of describing phenotypes at a broad level. Maybe the authors should be allowed to advocate on this issue

We agree with the reviewer's comment that we have not provided a molecular mechanism explaining how *Crossfire*, in combination with *Firre*, affects autosomal gene expression *in trans*. Strong evidence that the *Firre* lncRNA product functions *in trans* was recently provided by rescuing the hematopoiesis defects observed in *Firre* knockout mice, both at the physiological and molecular level, by activating the *Firre* transgene on a different chromosome. Therefore, the pronounced autosomal effect observed in the *Crossfire* and *Firre* double deletion could be due to a *Crossfire* RNA *trans*-mediated mechanism, as proposed for the *Firre* lncRNA, or indirectly by controlling *Firre* expression through RNA- and DNA-mediated mechanisms as well as the act of transcription. Furthermore, we cannot exclude the possibility that Xi-specific, distinct epigenetic features, including the conserved megastructures and open chromatin sites, may directly or indirectly impact autosomal gene regulation.

While we agree with the reviewer that these are very relevant questions that need to be addressed in the future, we believe that our comprehensive phenotypic investigation provides a starting point for further exploration of these non-coding loci *in vivo*. This includes generating one of the largest cohorts of genetically engineered mice targeting these X-linked non-coding loci individually and in combination, allowing us to address whether there are essential phenotypes. In addition, we performed a comprehensive multi-omic analysis that rules out a role for these loci in both imprinted and random X-chromosome inactivation (XCI) initiation and maintenance, the latter only possible through complex allele-specific single-cell analysis. Moreover, the comprehensive phenotyping approaches used in this study, including hundreds of measurements, are the first of their kind for X-linked long non-coding RNA loci allowing

us to shed light on the functional roles of these loci (**new Fig. 6**). Although the current study does not explore the detailed mechanisms by which these loci control autosomal genes, the comprehensive analysis and phenotypic insights provide a solid framework for future studies to explore the mechanisms of action of these non-coding loci under different conditions and lineages, which is beyond the scope of this study.

2. There is a seeming lack of understanding of the literature or overlooked results of major importance in the field. This is especially problematic given that it affects the first sentence in the manuscript abstract, namely: "The lncRNA *Crossfirre* was identified as the only imprinted X-gene" This statement is incorrect. *Xist*, and other genes in the X-inactivation centre, *Ftx* and *Jpx*, are notable examples, as are *Rhox5* (2006, PMID: 16431368) and *Fthl17* (PMID: 20185572). It might have been that the authors were referring specifically to imprinted X-genes expressed on the maternally inherited X chromosome, but even in that case, some genes have already been reported, e.g., in 2005 *Xlr3b* (PMID: 15908950), *Xlr4b* and *4c* (PMID: 15908953)

Thank you for bringing this matter to our attention. We are aware that *Xist* and other genes within the X-inactivation center show imprinted expression in early development and extraembryonic lineage, resulting in imprinted XCI, leading to imprinted maternal expression of the majority of X-linked genes. Similarly, we are also aware of the existence of the mentioned X-linked genes, as we discussed and evaluated the imprinted status of these genes in our previous large-scale imprinted mapping study, where we examined imprinted expression in multiple organs during mouse development (PMID: 28806168 see Fig. 4c). In the mentioned study, we did not detect imprinted expression of the genes *Xlr3b*, *Xlr4b*, *Xlr4c*, *Fthl17*, and *Rhox5* due to insufficient SNP coverage or due to low or no expression. The only imprinted gene we detected outside imprinted XCI in somatic tissues was *Crossfirre*, which was detected by RNA-seq in the brain and validated with maternal H3K4me3 at the promoter.

The statement "The lncRNA *Crossfirre* was identified as the only imprinted X-linked gene" was thus in the context of our recent comprehensive imprinting mapping study. However, we agree with the reviewer's concern that this statement is misleading without context, and have revised the sentence accordingly.

We clarified this matter in the introduction of the manuscript and noted the maternal imprint of *Xist* in the Introduction [line number: 90]: "An additional X-linked lncRNA, *Crossfirre* (*Gm35612*), is transcribed antisense to the *Firre* lncRNA. *Crossfirre* consists of 3 exons and its 3' end is located 500bp from the 3' end of *Firre*. In a comprehensive allele-specific analysis, *Crossfirre* was identified as an imprinted lncRNA in somatic tissues, predominantly transcribed from the maternal allele. Since *Crossfirre* expression marks the maternal X chromosome, this locus may warrant further investigation for a link to imprinted XCI in addition to the maternal imprint controlling the *Xist* locus".

3. Overall, the flow of the manuscript lacks clarity and coherence. The reviewer found it challenging to follow the logical trial of the text and observed a disjointed progression from one experiment to another. Lacking citations and correspondence with previous accounts in the literature has also contributed to this (see point 1). The figures are beautifully presented, but are often not clear to understand and interpret.

We acknowledge the reviewer's comment. In response, we have made corrections throughout the manuscript, including the Introduction, Results, and Discussion sections, and added the necessary citations. In particular, we have focused on improving clarity by refining ambiguous sentences, as well as improving key figures (see **Fig. 3**, **Fig. 4**, **Fig. 6**) and expanding Supplementary Figures (see **Supplementary Fig. 1**, **Supplementary Fig. 3**, **Supplementary Fig. 4**, **Supplementary Fig. 5**, **Supplementary Fig. 6**). We further detailed our **Supplementary Table** to include information on phenotype effect sizes and expression values (**Supplementary Table o-s**). In addition, we have performed new analyses and experiments (see **Supplementary Fig. 1**, **Supplementary Fig. 3**, **Supplementary Fig. 4**, **Supplementary Fig. 5**, **Fig. 6**, **Supplementary Fig. 6**) to address the remaining questions and to strengthen the coherence of the manuscript. Overall, we are confident that these revisions have significantly improved the manuscript.

Minor points:

Unclear sentences:

L86: "Has been identified in a recent annotation" Please provide a reference.

We agree with the reviewer's comment and have added the reference of the Refseq annotation that, to our knowledge, first reported *Crossfirre* (*Gm35612*, downloaded in February 2018). For clarity, we have removed the sentence and included this information in the Methods section.

Edited in the Methods section of the manuscript [line number: 567] "For the allele-specific RNA-seq analysis, we used the GRCm38/mm10 RefSeq gene annotation (**PMID: 26553804**, downloaded in February 2018) in conjunction with the previously described SNP dataset (PMID: 31738164) containing 15,438,314 SNPs distinguishing CAST and BL6 strains (PMID: 21921910)."

L92: Why would a gene expressed specifically on the maternal X warrants further investigation for a link to XCI? For the reviewer this is not obvious. Please explain.

Since *Crossfirre* is specifically expressed on the maternal X chromosome, we speculate that this locus may serve as a marker for imprinted XCI in addition to the maternal imprint controlling the *Xist* locus, and thus warrant further investigation to link to imprinted XCI.

We clarified this matter in the introduction of the manuscript [line number: 92] "In a comprehensive allele-specific analysis, *Crossfirre* was identified as an imprinted lncRNA in somatic tissues, predominantly transcribed from the maternal allele. Since *Crossfirre* expression marks the maternal X chromosome, this locus may warrant further investigation for a link to imprinted XCI in addition to the maternal imprint controlling the *Xist* locus."

L100-104: why were the authors expecting more of an X-specific phenotype?

We appreciate the reviewer's request for greater clarity, and we did not intend to imply an expectation of results. We now revised the sentence to emphasize that even though the investigated loci show distinct epigenetic features on the X chromosome, including

the Xi-specific conserved megastructures and open chromatin as well as imprinted expression of *Crossfirre*, their absence did not affect XCI biology.

We clarified this matter in the introduction of the manuscript [line number: 107]: “Despite their distinct epigenetic features observed on the X chromosome, including the Xi-specific conserved megastructures and open chromatin as well as imprinted expression of *Crossfirre*, our extensive multi-omics investigation uncovered an interplay between *Crossfirre* and *Firre* in autosomal gene regulation, rather than affecting XCI biology.”

L215: “We detected a more pronounced skewing ratio in the TKO heterozygous spleen cell population” The reviewer wants to know if this finding is significant. If so, it should be interesting to discuss and investigate.

Thank you for pointing this out. To determine the significance of the pronounced skewing ratio in the TKO heterozygous spleen cells, we performed the Fisher’s exact test. The test was based on the following cell counts: WT (Cast Xa: 1342, BL6 Xa: 640) and TKO (Δ Xi: 1359, Δ Xa: 243), and it revealed a significant result (p -value: $2.772737e-33$; Odds ratio: 0.375).

However, it is important to note that the skewing of the XCI ratio is established during the initiation of XCI by the lncRNA *Xist*. Consequently, the skewing ratio is reflected in the allelic ratio of *Xist*, as demonstrated in the **new Supplementary Fig. 4c** (see below). Thus, the significance in the skewing ratio observed in the single-cell data is based on a single biological replicate in the WT and heterozygous spleens.

To verify this result, we repeated the experiment using the same breeding system (TKO $-/+$ x CAST) and performed bulk RNA sequencing on three WT and three heterozygous spleens (**new Supplementary Fig. 4d**, see below). Then we used Allelome.PRO to calculate the allelic ratios for the X-linked genes and observed the expected skewing ratios in F1 CASTxBL6 crosses for the WT samples that anti-correlate with the *Xist* ratios. In contrast to the observation of the single-cell data, we observed a similar pattern also for the heterozygous TKO samples (**new Supplementary Fig. 4e**, see below). To test for significance, we computed a t-test between the allelic ratios of *Xist* for WT and mutant and did not observe any significant differences (**new Supplementary Fig. 4f**, see below). This observation indicates that the more pronounced skewing ratio seen in the TKO heterozygous single-cell spleen data is likely not a result of the TKO, but rather reflects individual-specific variation of the skewing ratio.

We have added the following text to the Results section of the manuscript [line number: 228]: “Notably, we detected a more pronounced skewing ratio in the TKO heterozygous spleen cell population (82.8% TKO on Xi and 14.8% TKO on Xa, **Fig. 4c, Supplementary Fig. 4b-c**). To verify the heterozygous TKO has an impact on the skewing ratio, we replicated the experiment using bulk RNA sequencing on WT ($n = 3$) and $-/+$ TKO ($n = 3$) spleens (**Supplementary Fig. 4d**). In contrast to the results of the single-cell data, we observed a comparable skewing ratio between WT and heterozygous TKO samples, with no significant differences observed for the *Xist* allelic ratio (**Supplementary Fig. 4e-f**).”

The new **Supplementary Fig. 4** is provided below for your convenience.

Supplementary Fig. 4: Allele-specific single-cell and bulk RNA-seq analysis of F1 spleens.

a, Violin plots displaying the allelic ratio for each autosome from single-cells of wildtype (WT, gray) and heterozygous Δ *Crossfirre-Firre-Dxz4* (TKO, turquoise) spleens in females. **b**, Violin plot of allelic ratios for the X chromosome per cell in WT (gray) and heterozygous TKO (turquoise) spleens. Allelic ratios range from 0 to 1, where 0 corresponds to 100% BL6 Xa and 1 corresponds to 100% CAST Xa. Allelic ratios between 0.3 to 0.7 were classified as biallelic, highlighting cells with both X chromosomes active. **c**, Violin plot of allelic ratios for the X-linked genes. Single-cell reads were combined as pseudobulk for WT and heterozygous TKO samples. For box plots, the center line marks the median, and the box represents the interquartile range. Whiskers extend to the minimum and maximum values within 1.5 times the interquartile range. **d**, Schematic workflow showing the experimental setup to further investigate the X chromosome inactivation skewing ratio in WT and heterozygous TKO samples. Heterozygous TKO females (BL6) were mated with WT CAST males to generate F1 hybrids with WT and heterozygous TKO genotypes. Spleens were isolated ($n = 6$) and processed for bulk RNA-seq. **e**, Violin plots showing the allelic ratios of the X-linked genes for each replicate (WT $n = 3$, +/- TKO $n = 3$). **f**, Allelic ratios plotted for *Xist* per replicate (WT $n = 3$, +/- TKO $n = 3$). A t-test was used to assess significance between the allelic ratios of WT and heterozygous TKO mutants. Whiskers range from minimum to maximum values.

L371: "This study found that *Firre* RNA expressed from the Xa chromosome maintains histone H3K27me3 enrichment on the Xi without reactivating X linked genes". For the reviewer is this an unclear sentence.

We thank the reviewer for pointing this out. What we meant to say with this sentence was "This study found that *Firre* RNA expressed from the Xa acts *in trans* by maintaining histone H3K27me3 enrichment on the Xi". However, after restructuring the discussion, we felt that this statement was no longer necessary to follow the logic and thus removed this sentence from the discussion.

2. Unclear figures and analyses:

Figure 1b and c. The axis legend: Log10(p) is wrong, in both b and c as well as in the material and method, as the authors clearly plotted "1-log10(p)". The analysis,

particularly regarding how the sliding window is defined, needs to be more thoroughly explained.

We appreciate the reviewer's comment and would like to clarify that we plotted the \log_{10} of the p -value based on the binomial distribution, assuming an equal number of peaks between the sexes. To represent the difference in the number of peaks between females and males, we assigned positive values when females had more peaks and negative values when this was the case for males. We have detailed this approach in the Methods section to ensure clarity for the reader. Furthermore, we extended our explanation of the sliding windows.

We clarified this matter in the methods section of the manuscript [line number: 530]:
“To assess differences in the chromatin profile of male and female samples, we intersected broad peaks called by MACS2 for each organ and sex. We then counted the number of peaks within sliding windows of 100Kb across the entire genome. The intervals between the windows were 50Kb. The median number of peaks per window was calculated across organs and sex and used to calculate \log_{10} p -values from a binomial test to assess significant differences. A positive value was assigned if more peaks were present in females than in males, while negative values were assigned if the reverse was true.”

Furthermore, previous studies (ref.11 in the paper) have already demonstrated that *Firre* is the only domain producing accessibility peaks on the Xi in ATAC-seq, and this finding should be addressed in the discussion.

We thank the reviewer for their comment. In response, we have added a sentence to the Discussion referencing key literature that previously observed Xi-specific sites in F1 cell lines and selected organs [line number: 345]:

“Our study confirmed the previously observed female-specific chromatin accessibility of *Crossfirre*, *Firre* and *Dxz4*, known to originate from Xi (PMID: 27437574, PMID: 28806168, PMID: 30479398, PMID: 25887447). Remarkably, we identified this pattern as the most female-specific epigenetic signature genome-wide, a finding that was consistent across organs.”

Our contribution in the current study involved analyzing ATAC-seq data across a comprehensive set of male and female organs and conducting a genome-wide analysis. This approach enabled us to demonstrate that these loci contain the most significant sex-specific regions on a genome-wide scale.

Figure 2.d 50 pups from 8 litters are 6.25 pups per litter, 55 are 6.875, is there a reason why 6.25 is rounded to 6.3 but 6.875 is rounded to 6.8. This seems incoherent, please explain this.

Thank you very much for this observation. We revised the values in this table by rounding consistently. 6.875 is now rounded up to 6.9. This also affected the p -value of Δ *Crossfirre*, changing from 0.33 to 0.34. Other values in this table remain the same. The updated table is provided below for your convenience:

d

Mating GT [-/- x -/y]	Litter	Pups	Litter size	♂	♀	p -value	Sex ratio
Δ Crossfirre	8	50	6.3	27	23	0.34	117.4
Δ Crossfirre-Firre	8	55	6.9	28	27	0.5	103.7
TKO	5	47	9.4	23	24	0.61	95.8

Fig. 2: Mice carrying a *Crossfirre* single deletion or combined with *Firre* and *Dxz4* are viable and undergo normal development.

d, Sex distribution of homozygous Δ *Crossfirre*, Δ *Crossfirre-Firre*, and TKO breeding. The *p*-values are obtained from a binomial test, assuming an equal number of peaks between the sexes.

Figure 3d. The reviewer finds it concerning that both controls (Xi and Xa deletion, but without deletion) show different allelic ratios violin plots. This cast doubts on the reproducibility of the author's allelic ratio determination. This should at least be discussed.

We appreciate the reviewer's comment regarding the reproducibility of allelic ratio determination. The slightly different allelic ratios seen in the violin plots are due to the varying genetic backgrounds of the CAST and BL6 X chromosomes. More specifically, because of imprinted XCI, the CAST chromosome is active in the forward cross (CAST mother and BL6 father), and the BL6 X chromosome is active in the reverse cross (BL6 mother x CAST father). In the most extreme cases, this results in strain-specific escape, a well-known phenomenon (PMID: 28134930, PMID: 28806168, PMID: 34752748). To clarify this and demonstrate the reproducibility of our allele-specific quantification, we have included a **new Supplementary Fig. 3**. This figure shows the allelic ratios of X-linked genes for wildtype replicates of both the forward and reverse crosses ($n = 17$), illustrating the consistency of the violin plots for each cross and the difference in the number of escapees, with a higher frequency of CAST-specific escape in the reverse cross (**Supplementary Fig. 3a**). We further emphasize the strain-specific escape genes in the heatmap of the **new Supplementary Fig. 3b**, many of which were recently reported in the placenta (PMID: 31738164). We have included this information in the Results section of the manuscript [line number: 195]. The new **Supplementary Fig. 3** is provided below for your convenience:

Figure 3e. It would be helpful if the authors could provide a list of genes they are plotting to facilitate further analysis of the graph. Additionally, the absence of *Firre*/*Crossfire* from the BL6 strain in the Xa-specific plot is not discussed. If *Crossfire* is maternally imprinted, its absence should result in no allelic bias in the plots when the deletion is on the Xa. This also applies for the *Dxz4* KO.

We thank the reviewer for this suggestion and have updated **Fig. 3e** to include the gene names in the heatmap. Furthermore, we would like to mention that the absence of *Crossfirre*, *Firre* and *Dxz4* from the BL6 strain in the Xa-specific plot is due to the fact that the heatmap only shows genes that are informative in all samples. This is the reason why *Firre* and the other loci are not shown, as they are completely abolished in the Xa deletion. The arrows in the heatmap indicate only the approximate positions of these loci, allowing us to investigate whether there is a putative *cis*-effect in the

absence of these loci in Xi or Xa, which we did not observe (see also **Supplementary Fig. 3c**). For enhanced clarity, we have included this explanation in the figure legend. The **updated Fig. 3** is provided below for your convenience:

Fig. 3: Deleting the imprinted *Crossfirre* locus alone or together with *Firre* and *Dxz4* does not affect imprinted XCI.

a, Simplified schematic of our experimental system to investigate the impact of the deletions on the inactive X (Xi, left) or active X (Xa, right) for imprinted X inactivation. E12.5 female placentas are isolated from wildtype (WT) F1 reciprocal crosses ($n = 17$; 9 CASTxBL6, 8 BL6xCAST background), and for the six F1 mutants carrying the paternally inherited deletion on Xi ($n = 3$ for each genotype) or the maternally inherited deletion on Xa ($n = 3$ for each genotype) and subjected to RNA-seq. The relative expression (mean and standard deviation) between WT and Δ *Crossfirre-Firre-Dxz4* (TKO, turquoise) is shown for female placentas carrying the deletion on Xi (left) and Xa (right). **b**, The number of differentially expressed genes in the placenta is shown below each knockout strain for the maternal or paternal deletions. **c**, Volcano plot showing differentially expressed genes between WT and TKO on Xi (left) and Xa (right). The Venn diagram highlights the low overlap of dysregulated genes between the double deletions Δ *Dxz4-Firre* (blue) and Δ *Crossfirre-Firre* (green) and the TKO (turquoise). **d**, Violin plots showing median allelic ratios for X-linked genes in WT (black) and the six knockout strains carrying the deletions on Xi (left) or Xa (right). The blue dot emphasizes the paternal allelic ratio of the lncRNA *Xist*. The allelic ratios range from 0 to 1 such that 1 corresponds to 100% expression from the maternal allele (MAT, red), 0.5 to biallelic expression, and 0 to paternal expression (PAT, blue). Consequently, the allelic ratio of BL6xCAST samples was adjusted by subtracting the ratio from 1. For box plots, the center line marks the median, and the box represents the interquartile range. Whiskers extend to the minimum and maximum values within 1.5 times the interquartile range. Points outside this range are defined as outliers. **e**, Heatmap showing median allelic ratios for X-linked genes that are informative across all samples in WT and the six knockout strains carrying the deletions on Xi (upper panel) or Xa (lower panel). The brown color indicates an allelic ratio of 1 corresponding to the CAST allele, while black indicates an allelic ratio of 0 (BL6 allele). We highlighted two common escape genes *Kdm6a* and *Eif2s3x* with biallelic expression, thus validating our approach. Arrows indicate the approximate location of *Crossfirre*, *Firre*, and *Dxz4*. *The expression of *Tsix* from Xi is due to the overlapping nature with *Xist* and thus an artifact of the non-stranded sequencing procedure.

Figure 4 analysis. The numbers do not add up: WT = 2046 but 1342+640+61=2043
Heterozygous = 1648 but 1359+243+40=1642.

We appreciate the reviewer's accurate observation. The number of cells after quality control was 2046 for WT and 1648 for TKO samples. However, for the allele-specific analysis, we obtained informative allele-specific transcriptomes for 2043 (WT) and 1642 (TKO) cells. We agree that this is misleading in the manuscript. Therefore, we have revised the sentence accordingly.

We clarified this matter in the introduction of the manuscript [line number: 216] "After quality control and normalization, we obtained the allele-specific single-cell transcriptome for 2043 WT and 1642 heterozygous cells (see **methods**)."

Figure 4e. There seems to be a similar issue as in Figure 3. If *Dxz4/Firre* and *Crossfire* are deleted from the active (Xa) BL6 chromosome, why are all reads for these genes depicted in black? If the triple knockout still shows expression from all genes on the active X chromosome, it raises concerns.

We appreciate the reviewer's comment. As mentioned in our response to Fig. 3e, the absence of *Crossfirre*, *Firre*, and *Dxz4* from the Xa-specific plot is due to the fact that the heatmap only shows genes that are informative in all samples. Therefore, the arrows indicating *Crossfirre*, *Firre*, and *Dxz4* only represent the approximate positions, allowing us to explore potential *cis*-effects in the absence of these loci on the Xi or Xa chromosome. To improve clarity, we have included this explanation in the legend of **Fig. 4d**.

For enhanced clarity, we have included this explanation in the figure legend: "Arrows indicate the approximate location of *Crossfirre*, *Firre*, and *Dxz4*"

Figure 6. Difficult to interpret. The reviewer is unsure about the significance of a 1 Cohen's d unit and what the absence of an effect indicates. In correspondence to the phenotype analysis in figure 6. It would be beneficial to explore any interesting phenotypes further, especially if they are reproducible and significant. It could be insightful to conduct more analyses beyond the hematopoietic cell lineage, especially considering this has already been addressed in a previous publication from the lab. In addition, investigating how these findings correlate with single-cell analysis of organs could provide interesting insights.

We sincerely appreciate the reviewer's feedback regarding **Fig. 6**. In response to concerns about interpretability, we have refined the presentation of our phenotypic results in the manuscript. The new **Fig. 6** provides a comprehensive overview of the phenotyping results, including visualization of p -values, as well as effect size measurements, to facilitate the interpretation of significance. The selected parameters provide a global overview of the most important tests relevant for a first impression (**Fig. 6b**). In addition, we have summarized the significant parameters as phenotypes in a concise overview in **Fig. 6c**. The **new Fig. 6** is provided below for your convenience. In addition, a summary of the phenotyping results will be uploaded upon publication on the following webpage, providing additional information about each parameter test of the phenotyping pipeline:

<https://www.mouseclinic.de/results/phenomap-and-results/index.html>

Furthermore, we investigated how the phenotyping results correlate with the single-cell data from the spleen to strengthen the conclusion of the immune cell analysis. This matter was addressed in **reviewer 3, comment 5** (see **new Supplementary Fig. 6**).

Fig. 6: Large-scale phenotyping analysis of TKO mutants uncovers knockout and sex-specific phenotypes.

a, The German Mouse Clinic (GMC) phenotyping pipeline was conducted using 30 wildtype (WT, 15 males, 15 females) and 26 Δ *Crossfirre-Firre-Dxz4* (TKO) mice (13 males, 13 females). The pipeline covered the screening tests from the following categories: immunology/allergy, behavior, biomarkers, cardiovascular, clinical chemistry, pathology, dysmorphology, eyes, metabolism, neurology, and nociception. **b**, Visualization of key measured parameters to provide a general overview of the GMC screening pipeline results. The size of each triangle corresponds to the absolute effect size, represented by Cohen's d. Triangles pointing up or down indicate upregulation or downregulation, respectively. Parameters with a *p*-value < 0.05 are considered significant. N.S.: not significant (*t*-test). Additional parameter information, including *p*-values, effect sizes (Cohen's d, Hedges' g), and parameter abbreviations are provided in **Supplementary Table 1**, sheets o-q. **c**, Concise overview of the significant parameters as phenotypes for the groups TKO (*n* = 9), female-specific (*n* = 6) and male-specific (*n* = 13) by phenotyping category.

Reviewer #2

Hasenbein et al. present a comprehensive analysis of three intriguing X-linked lncRNA loci, two of which produce convergently transcribed RNAs, Crossfire and Firre, and a third which is also transcribed into a lncRNA and like the Firre locus harbors a large number of embedded DNA regulatory elements. The major findings of the study are significant to both the lncRNA and X-chromosome inactivation fields. Despite the distinct epigenetic features associated with the Crossfire, Firre, and DX4 loci, and the presence of these loci on the X chromosome, their combinatorial deletion results in no obvious defect in XCI. Intriguingly, however, tissue-specific phenotypes are observed on autosomal genes. The study is clearly presented and very well controlled. I found the methodology to be sound, that the conclusions were supported by the data, and that the methods were sufficiently detailed so that others could reproduce the analyses.

I noted that the authors referenced Supplementary Table 1 throughout the manuscript but I was unable to access this as a reviewer. My main request would be that the authors include in that table or another one like it the lists of differentially expressed genes, and their associated expression values in the tissues analyzed, if those data have not already been included.

We appreciate the reviewer's valuable feedback. We regret that the reviewer was unable to access **Supplementary Table 1**. The table included lists of differentially expressed genes, their log2 fold change, and their p-value in the tissues analyzed. We have updated **Supplementary Table 1** to also include the expression values as transcripts per million (TPM) for the bodymap (**Supplementary Table 1, sheet r**) and the placenta (**Supplementary Table 1, sheet s**). For your convenience, a section from the table is shown below:

	A	B	C	D	E	F	G	H	I	J	K	L	M	N	O	P	Q	R
1	Chromosome	Mgi_symbol	Start_position	End_position	Strand	Ensembl_gene_id	Transcript_length	Brain_WT	Brain_FD	Brain_TKO	Heart_WT	Heart_FD	Heart_TKO	Kidney_WT	Kidney_FD	Kidney_TKO	Liver_WT	Liver_FD
2	1	4933401J01Rik	3073253	3074322	+	ENSMUSG00000102693	1070	0	0,119	0	0	0	0	0	0	0	0	0
3	1	Gm126206	3102016	3102125	+	ENSMUSG00000064842	110	0	0	0	0	0	0	0	0	0	0	0
4	1	Nkx4	3205901	3671498	-	ENSMUSG00000051951	6094	35,465	29,012	27,879	0,1	0,095	0,094	0,051	0,052	0,039	0,005	0
5	1	Gm18956	3252757	3253236	+	ENSMUSG00000102851	480	0,36	0	0	0	0	0	0	0	0	0	0
6	1	Gm37180	3365731	3368549	-	ENSMUSG00000103377	2819	0	0,112	0,073	0	0	0	0	0	0,016	0	0
7	1	Gm37363	3375556	3377788	-	ENSMUSG00000104017	2233	0	0,033	0,047	0	0	0	0	0	0	0	0
8	1	Gm37686	3464977	3467285	-	ENSMUSG00000103025	2309	0,028	0,028	0,039	0	0	0	0	0	0	0	0
9	1	Gm1992	3466587	3513553	+	ENSMUSG00000089699	250	0	0	0	0	0	0	0	0	0	0	0
10	1	Gm37329	3512451	3514507	-	ENSMUSG00000103201	2057	0	0,031	0	0	0	0	0	0	0	0	0
11	1	Gm37341	3531795	3532720	+	ENSMUSG00000103147	926	0	0	0	0	0	0	0	0	0	0	0
12	1	Gm38148	3592892	3595903	-	ENSMUSG00000103161	3012	0	0,042	0,073	0	0	0	0	0	0	0	0
13	1	Gm19938	3647309	3658904	-	ENSMUSG00000102331	3259	0,856	0,549	0,3	0	0	0	0	0	0	0,019	0
14	1	Gm10568	3680155	3681788	+	ENSMUSG00000102348	1634	0,821	0,568	1,098	0	0	0	0	0	0	0	0
15	1	Gm38385	3752010	3754360	+	ENSMUSG00000102592	2351	0	0	0	0	0	0	0	0	0	0	0
16	1	Gm37396	3783876	3783933	-	ENSMUSG00000088333	58	0	0	0	0	0	0	0	0	0	0	0
17	1	Gm37381	3905739	3986215	-	ENSMUSG00000102343	1364	0	0,054	0	0	0	0,02	0	0	0	0	0
18	1	Rpl1	3999557	4409241	-	ENSMUSG00000025900	12311	0,056	0,048	0,051	0,917	0,922	1,372	0,496	0,486	0,532	0,003	0
19	1	Gm5101	4256234	4260519	-	ENSMUSG00000102948	1867	0	0	0	0	0	0	0	0	0	0	0
20	1	Gm37483	4363346	4364829	-	ENSMUSG00000104123	1484	0	0	0	0	0	0	0	0	0	0	0
21	1	Sox17	4490931	4497354	-	ENSMUSG00000025902	4772	4,107	3,915	4,393	7,092	5,532	6,536	7,46	8,005	8,442	1,541	0
22	1	Gm37587	4496551	4499558	+	ENSMUSG00000104238	2917	0,08	0,094	0,062	0,135	0,077	0,049	0,06	0,057	0,092	0,022	0
23	1	Gm37357	4522905	4526737	+	ENSMUSG00000102269	2991	0,182	0,085	0,239	0,289	0,149	0,284	0,294	0,358	0,432	0,134	0
24	1	Gm22307	4529017	4529123	+	ENSMUSG00000096126	107	0,621	0,594	0	0	0,959	0,675	1,902	0	0,57	0,419	0
25	1	Gm38076	4534837	4535286	-	ENSMUSG00000103003	450	0,248	0	0	0	0	0	0	0	0	0	0
26	1	Gm37323	4583129	4586252	-	ENSMUSG00000104328	2773	0	0,049	0,075	0,015	0,096	0,051	0,084	0	0,066	0,016	0
27	1	Gm37369	4610471	4611406	+	ENSMUSG00000102735	936	0	0	0	0	0	0	0	0,048	0,065	0	0
28	1	Gm5085	4687934	4689403	-	ENSMUSG00000098104	1470	0,09	0,173	0,255	0,099	0,245	0,05	0,268	0,09	0,053	0,187	0
29	1	Gm5119	4692219	4693424	-	ENSMUSG00000102175	1206	0	0,182	0,131	0,046	0,058	0,086	0,062	0	0,034	0	0
30	1	Gm25493	4723277	4723379	-	ENSMUSG00000088000	103	0	0	0,014	0	0	0	0	0	0	0,592	0
31	1	Gm2053	4735046	4735676	-	ENSMUSG00000103265	631	0,281	0	0	0	0	0	0	0,045	0,067	0	0

Beyond a desire for well annotated lists of differentially expressed genes (which may already exist in Table S1), I have very little to suggest. If the authors felt it were appropriate to do so, in the discussion section they could speculate as to whether Crossfire and the other lncRNAs function as RNA products or if it is the act of their transcription that is more likely to be the result of the mutant phenotypes upon their deletion.

We agree with the reviewer's comment that it is important to discuss the possible molecular mechanism explaining how *Crossfirre* in combination with *Firre* affects autosomal gene expression *in trans*, and have added the following section to the discussion [line number: 398]:

"In order to regulate autosomal genes, an X-linked lncRNA would be required to function *in trans*. This is consistent with the *trans*-acting role of the *Firre* RNA, proposed to convey its function through *trans*-chromosomal associations. Therefore, the more pronounced autosomal effect observed in the *Crossfirre-Firre* double deletion could be due to an RNA-mediated mechanism of *Crossfirre* itself or indirectly through the regulation of *Firre* via RNA- and DNA-mediated mechanisms or through the act of transcription. However, we did not observe a correlation between the expression levels of *Crossfirre* and *Firre* across organs, which may exclude a co-regulatory role in the investigated organs. Nevertheless, the mechanism underlying the observed autosomal dysregulation, and whether it is direct or indirect, remains to be elucidated"

Reviewer #3

The findings presented in this manuscript are very relevant for the non-coding genome research field as the authors underwent a detailed phenotyping at the molecular and macroscopic scales for non-coding transcribed loci and provide a robust pipeline to understand the functions of the non-coding genome. Moreover, the findings are relevant for the XCI field as the authors provide a clear demonstration of the absence of regulatory implication of *Crossfire*, *Firre* and *Dxz4* in the regulation of the mouse XCI, a lasting question in the field. Publication of this study is recommended on the basis of these important demonstrations. We however have some doubts on the structure of the manuscript. As XCI aficionados, we were more compelled by the non-XCI related phenotypes and left hanging with a sense that links/analysis/discussions were missing between observed gene deregulations and cell/organ phenotypes. The comments hereafter further elaborates on this.

Comments:

1. The title “Dissecting the in vivo role of X-linked lncRNA loci with conserved sex- and allele-specific epigenetic signatures” is too vague and does not reflect the results nor the conclusion of the manuscript. For instance, the conservation is not the topic of the manuscript moreover the sex and allele-specific epigenetic signatures of the loci under scrutiny are of very moderate interest regarding the authors findings. We feel that the title should name the loci under scrutiny, their trans acting role in autosomal gene regulation and their impact at the phenotypic level.

We agree with the reviewer that our proposed title may be too vague and should mention the X-linked loci, their *trans*-acting role in autosomal gene regulation, as well as their phenotypic impact. Therefore, we have replaced the original title with the following:

“X-linked deletion of *Crossfirre*, *Firre*, and *Dxz4* in vivo uncovers diverse phenotypes and combinatorial effects on autosomal gene regulation”

2. The authors often refer to *Crossfire* as the “only imprinted X-linked gene” or “*Crossfirre*, the only imprinted lncRNA located on the X chromosome”. Care should be given in revising these statements; *Xist*, an X-linked gene, also has imprinted regulation, albeit paternally and not in the soma.

Thank you for bringing this to our attention. We are aware of the existence of other imprinted X-linked genes, as discussed in detail in response to **reviewer 1 comment 2**. In our previous large-scale imprinted mapping study, we examined imprinted expression in multiple organs during mouse development (PMID: 28806168). In that study, the only imprinted gene we detected in somatic tissues was *Crossfirre*, which was detected by RNA-seq and validated by maternal H3K4me3 at the promoter. The statement “The lncRNA *Crossfirre* was identified as the only imprinted X-linked gene” was thus in the context of our recent comprehensive imprinting mapping study. However, we agree with the reviewer's concern that this statement is misleading without this context, and have revised the sentence accordingly.

We clarified this matter in the introduction of the manuscript and noted the maternal imprint of *Xist* in the Introduction [line number: 90]: “An additional X-linked lncRNA, *Crossfirre* (*Gm35612*), is transcribed antisense to the *Firre* lncRNA. *Crossfirre* consists

of 3 exons and its 3' end is located 500bp from the 3' end of *Firre*. In a comprehensive allele-specific analysis, *Crossfirre* was identified as an imprinted lncRNA in somatic tissues, predominantly transcribed from the maternal allele. Since *Crossfirre* expression marks the maternal X chromosome, this locus may warrant further investigation for a link to imprinted XCI in addition to the maternal imprint controlling the *Xist* locus".

3. The authors write "we speculate that the locus [Crossfire] may serve as a marker to prevent silencing of the maternal X chromosome in extraembryonic lineages with imprinted XCI": maternal imprint of the *Xist* promoter has been fairly well established and as it is formulated, this hypothesis seems unlikely and does not need to be stated as is. Moreover, the authors and others previously shown that the *Firre-Dxz4* loci is not regulating imprinted nor random XCI in mice. Altogether this XCI narrative undermines the very interesting findings of the role *Crossfire-Firre-Dxz4* in autosomal transcriptional regulation and the thorough phenotyping study the authors have conducted. Hence, we would have favored a narrative centered around how X-linked loci can affect autosomal gene regulation and integrate their manuscript in a corpus of publications addressing those questions (Brenes 2021, Richart 2022, Roman 2024). We strongly feel the need to expand their analysis in the autosomal implication of the loci (see below).

We thank the reviewer for this comment. Regarding our hypothesis that *Crossfirre* may serve as a marker to prevent silencing of the maternal X chromosome in extraembryonic lineages with imprinted X chromosome inactivation (XCI), it is important to note that at the time we identified *Crossfirre* as a maternally expressed X-linked gene in somatic tissues and when the study was designed, the paper by Azusa Inoue et al. (PMID: 29089420) had not yet been published. We have noted the maternal imprint of *Xist* in the Introduction [line number: 92]:

"In a comprehensive allele-specific analysis, *Crossfirre* was identified as an imprinted lncRNA in somatic tissues, predominantly transcribed from the maternal allele. Since *Crossfirre* expression marks the maternal X chromosome, this locus may warrant further investigation for a link to imprinted XCI in addition to the maternal imprint controlling the *Xist* locus"

In addition, we agree that autosomal gene regulation by X-linked loci is an intriguing finding that warrants further investigation (see below), but we would like to emphasize the urgent need for this study to investigate the *in vivo* role of these loci in random and imprinted XCI, an ongoing question in the field. Our previous study showed that deletion of *Firre* and *Dxz4* does not affect XCI initiation, as knockout mice are viable, fertile, and have normal sex ratios. However, the precise effect on random XCI maintenance at the resolution of X-linked genes in adult organs could not be addressed previously due to the lack of an allele-specific single-cell analysis, required given the random nature of XCI. Using our single-cell approach combined with allele-specific single-cell sorting based on the XCI status, we were able to address this longstanding question. We showed that neither the absence of lncRNA expression nor the absence of Xi-specific chromatin structures affects XCI maintenance. Furthermore, the *in vivo* role of the imprinted lncRNA *Crossfirre* in the context of XCI biology was completely unexplored. Although we have not been able to assign a

function to these loci on the X chromosome, we believe that these findings are highly relevant to the XCI community.

However, we agree with the reviewer that the role of these loci in autosomal gene regulation is indeed an important finding. Therefore, we have expanded the analysis by correlating the expression of *Crossfirre*, *Firre*, and *Dxz4* with the phenotype intensity (see comment 4, **new Supplementary Fig. 5b**), as well as re-analyzing the scRNA-seq data to deepen the molecular exploration of the TKO phenotype on cell type populations (see comment 5, **new Supplemental Fig. 6**). In addition, we have detailed the TKO phenotyping by refining **Fig. 6** (see comment 6, **new Fig. 6**).

4. Having the expression levels of *Crossfirre*, *Firre* and *Dxz4* across the different tissues in WT samples could help connect the strong phenotype they observe in the spleen and potentially speculate on the molecular mechanism. Indeed, the trans-acting effect suggest that the function is mediated through the RNA molecule produced from those loci.

We thank the reviewer for this suggestion. To address this question, we first computed a Pearson correlation of expression levels (TPMs) between *Crossfirre*, *Firre*, and *Dxz4* in the six WT organs shown in **Fig. 1e**. We observed no significant correlation between these loci, suggesting that *Crossfirre*, *Firre*, and *Dxz4* do not co-regulate each other in the organs examined (**new Supplementary Fig. 1c**).

Supplementary Fig. 1: Imprinting of the *Crossfirre* locus and correlation analysis of *Crossfirre*, *Firre* and *Dxz4*.

a, Genome browser track of strand-specific RNA-seq data from female F1 brain covering the *Crossfirre* (Gm35612) and *Firre* locus¹⁵. The zoom out below shows the aligned forward sequencing reads of *Crossfirre*

after allele-specific splitting using SNPsplit⁵³. Sequencing reads originating from the FVB and CAST allele are indicated in black and brown, respectively. The cross scheme is depicted next to the browser track. **b**, Allele-specific splitting of H3K4me3 sequencing reads covering the *Crossfirre* promoter in female mouse embryonic fibroblasts (genetic background as in a). Sequencing reads originating from the FVB and CAST allele are indicated in black and brown, respectively. Publicly available data was used from ⁵⁴. **c**, Pearson's correlation of mean TPM values between *Crossfirre*, *Firre* and *Dxz4*.

In order to better understand how autosomal phenotypes are related across organs, we further plotted the wildtype expression of *Crossfirre*, *Firre*, and *Dxz4* ($\log_{10}(\text{TPM}+1)$) against the number of differentially expressed genes observed in the homozygous TKO bodymap. However, this analysis did not reveal any significant correlations between the expression levels and the number of differentially expressed genes for all three lncRNAs. The **new Supplementary Fig. 5b** is provided below for your convenience:

Supplementary Fig. 5: Quality control of the adult transcriptomic bodymap and downstream molecular analysis.

a, Pearson correlation heatmap of the different adult samples included in the bioinformatic analysis ($n = 76$). The correlation matrix is based on TPM values. **b**, Scatter plot showing the log₁₀-transformed mean TPM+1 correlation between *Crossfirre* (red), *Firre* (orange), and *Dxz4* (gray) and the number of significantly differentially expressed genes in Δ *Crossfirre-Firre-Dxz4* (TKO) samples across the studied tissues ($n = 6$). Correlations were calculated using Pearson's correlation coefficient. **c**, Dysregulated gene sets of TKO homozygous organs. Top 50 enriched dysregulated gene sets (FDR-adjusted p -value ≤ 0.1) for each bodymap organ of TKO females. The GSEA analysis was performed on DEseq2 test statistics with all gene ontology gene sets (c5.go.v7.4.symbols).

5. The authors should use their scRNA-Seq spleen data to deepen the molecular exploration of the TKO phenotypes they undergone with bulk RNA-Seq, what cell populations are the most affected by the TKO? How does it relate to the results of their phenotypic screen? Could it explain sex-specific effects?

As suggested by the reviewer, we have now used the single-cell spleen data to calculate the differences in cell type composition between cells carrying the TKO on Xa or Xi and the corresponding control (**new Supplementary Fig. 6**). We find that cells carrying the deletion on Xa, and thus lacking lncRNA expression, show a significant reduction in the proportion of B cells. This result is consistent with the phenotyping of the GMC, which observed a significant reduction of B cells in both males and females in whole blood, as well as with the proposed role of *Firre* in hematopoiesis (PMID: 31723143).

Interestingly, we also observed an effect on cell composition when the deletions were on Xi. In this setting, CD4+ T cells were significantly reduced, while B cells remained unaffected, suggesting an impact of the epigenetic signatures that are only present in females. While more work is needed to solidify these findings, the differential effects on cell type composition between cells lacking lncRNA expression that is present in both sexes and the lack of a female-specific epigenetic signature, provide an explanation for the sex-specific phenotypes observed in the phenotyping analysis. The **new Supplementary Fig. 6** is provided below for your convenience:

Supplemental Fig. 6: Cell type proportions from scRNA-seq data.

a, Barplot illustrating the distribution of cell types as a percentage derived from scRNA-seq cell counts from wildtype (WT, BL6 Xa) cells and heterozygous Δ *Crossfirre-Firre-Dxz4* (TKO) on Xa. Asterisks indicate statistically significant changes between WT and TKO samples using Fisher's exact test. The right panel shows the odds ratios obtained by Fisher's exact test for cell types containing more than 20 cells, with significant p -values highlighted in red. **b**, Same as in **a**, for WT (CAST Xa) cells and heterozygous TKO on Xi.

6. The phenotyping of the mice TKO mutants is for us the most compelling aspect of the paper and should be further exploited and described both in the main text and in the discussion

1. In the main text: put p-values associated to the Cohen's d values to help the reader interpret the figures. Moreover, we suggest to the authors to add another measure of the effect size, the Hedges' g, that is better powered for small sized cohorts. We believe the authors should try to connect the phenotype to the findings of their screen and the results of their multiple differential expression analysis. For example, how does TKO mice phenotypes relate to the enriched GO terms they are finding? We strongly believe that the analysis of the scRNA-Seq spleen data could strengthen the conclusions related to the Immune cell analysis.

We thank the reviewer for these suggestions. We now provide a comprehensive overview of the phenotyping results, including visualization of p -values, as well as effect size measurements, to facilitate the interpretation of significance. The selected parameters provide a global overview of the most important tests relevant for a first impression (**Fig. 6b**). In addition, we have summarized the significant parameters as phenotypes in a concise overview in **Fig. 6c**. In addition, a summary of the phenotyping results will be uploaded upon publication on the following webpage, providing additional information about each parameter test of the phenotyping pipeline:

<https://www.mouseclinic.de/results/phenomap-and-results/index.html>

Moreover, we implemented the reviewer's suggestion to include another measure of the effect size, the Hedges' g. The Hedges' g is a variation of Cohen's d. It is calculated in the same way based on the difference in means, but uses a corrected standard deviation in the denominator. Indeed, it can be more accurate for small sample sizes and has better properties when sample sizes are unbalanced. However, since our phenotypic analysis will be part of the large phenotyping resource of the German Mouse Clinic, which uses Cohen's d, we used the same statistic to be comparable with the other phenotyping studies. Therefore, we have kept the Cohen's d along with the p -values in the new **Fig. 6**. Nevertheless, to address the reviewer's suggestion, we also calculated Hedges' g for all of our comparisons and included the results in **Supplementary Table 1, sheet o-q**, and found that the effect size between Hedges' g and Cohen's d are very similar. The **new Fig. 6** is provided below for your convenience.

Fig. 6: Large-scale phenotyping analysis of TKO mutants uncovers knockout and sex-specific phenotypes.

a, The German Mouse Clinic (GMC) phenotyping pipeline was conducted using 30 wildtype (WT, 15 males, 15 females) and 26 Δ *Crossfirre-Firre-Dxz4* (TKO) mice (13 males, 13 females). The pipeline covered the screening tests from the following categories: immunology/allergy, behavior, biomarkers, cardiovascular, clinical chemistry, pathology, dysmorphology, eyes, metabolism, neurology, and nociception. **b**, Visualization of key measured parameters to provide a general overview of the GMC screening pipeline results. The size of each triangle corresponds to the absolute effect size, represented by Cohen's d. Triangles pointing up or down indicate upregulation or downregulation, respectively. Parameters with a p -value < 0.05 are considered significant. N.S.: not significant (t -test). Additional parameter information, including p -values, effect sizes (Cohen's d, Hedges' g), and parameter abbreviations are provided in **Supplementary Table 1**, sheets o-q. **c**, Concise overview of the significant parameters as phenotypes for the groups TKO ($n = 9$), female-specific ($n = 6$) and male-specific ($n = 13$) by phenotyping category.

In addition, we have added a paragraph in the Discussion where we relate the observed molecular phenotypes mediated by the combined deletion of *Crossfirre* and *Firre* to the identified phenotypes of the GMC [line number: 388]:

"Our transcriptomic analysis in the TKO revealed up-regulation of mitochondrial and ribosomal gene sets, suggesting a role in energy metabolism (PMID: 33092903, 35728540). Consistent with these expectations, we observed several phenotypes that may be related to these molecular findings. Among these are decreased urea levels and plasma cholesterol concentrations, which may be attributed to altered protein metabolism (PMID: 37118349). We also found decreased creatine levels and lactate concentrations with increased triglyceride levels, further supporting shifts in energy metabolism (PMID: 36050306, 32493980, 20304692)."

The request to use the single-cell spleen data to further strengthen the conclusions drawn from the immune cell analysis has already been addressed above (see **new Supplementary Figure 6**).

2. In the discussion: To our knowledge, the phenotyping approaches the authors have conducted is a first of its kind for non-coding elements and particularly lncRNAs. It would be nice to discuss in a couple of sentences how the results compare to similar studies working on protein-coding genes.

We appreciate the reviewer's comment highlighting the novelty of the extensive phenotypic screen on non-coding genes. In general, it is difficult to answer this question because the phenotypes of protein-coding gene knockouts are extremely variable. However, we can confirm that the effects on some parameters, such as lactate reduction in male TKO mice, are among the strongest detected by the GMC out of 95 mutants, with the majority of which are protein-coding knockouts. We have added the following sentence in the discussion to address the reviewer's comment [line number: 394]:

"Notably, the reduction in plasma lactate levels was one of the strongest measurements made by the GMC out of nearly one hundred knockout screens, including primarily protein-coding genes."

The authors should discuss the general and sex-specific phenotypes they found in light of the sex-specific epigenetic patterns they describe in their first figures. Particularly, they are stronger effect size (Cohen's $d \geq 1$) for males than females, yet the epigenetic patterns they described are mostly female-biased.

Indeed, the predominantly female-biased open chromatin pattern at these loci reflects their sex-specific properties. While it may seem intuitive to expect a prevalence of female-biased phenotypes, to our surprise we observe more phenotypes in males compared to females (male = 13, female = 6). Nevertheless, our results show a higher incidence of sex-specific ($n = 19$) compared to KO-specific phenotypes ($n = 9$), which is in line with expectations following deletion of the topmost sex-specific loci. As discussed above, the single-cell spleen results provide an explanation for the sex-specific phenotypes observed in the phenotyping analysis.

Minor comments

Throughout the text TKO is employed but, in the figures, Δ TKO, which doesn't make sense, is used. Please remove the Δ from the Δ TKO captions.

Thank you for bringing this to our attention. As suggested by the reviewer, we have removed the Δ from the Δ TKO captions.

SupF 1A: the RNA-seq track is not visually informative, a y-axis would help, or a simple count table. Y-axis would also be useful when looking at SupF 1B.

We agree with your suggestion and have incorporated Y-axes in all genome browser tracks throughout the manuscript (see **Fig. 1a**, **Supplementary Fig. 1a-b**, **Fig. 2b**).

Fig2 C: Please put the wt or Δ under the axis. A floating Δ is not necessary.

We have adjusted **Fig. 2c** according to the reviewer's suggestion.

Fig2D legend: Although we commend reproducible research, we do not feel inclusion of R formula for the determination of the p-value is appropriate.

We agree with this and have, accordingly, removed the formula from the figure legend.

Fig4A: Instead of RBC-, TER-119- would be more coherent with Zombie-

We agree with the reviewer's comment and adjusted the label accordingly (see **Fig. 4a**).

We sincerely thank the reviewers for their time and effort in evaluating our manuscript. Their insightful feedback was valuable. We have addressed the comments and provide point-by-point responses below. We believe that the revisions have substantially improved the manuscript.

Reviewer #1

The authors have done an excellent and thorough revision that have addressed all of our comments.

We thank the reviewer for the positive feedback and are pleased that our revisions have met the expectations. We appreciate the thoughtful comments during the revision process and believe they have greatly contributed to strengthening the manuscript.

Reviewer #2

The authors have addressed my prior requests and I would support publication of this study, which I believe describes results that are important for the XCI and lncRNA communities.

We thank the reviewer for the support and for recognizing the importance of our study to the XCI and lncRNA communities. We appreciate the constructive feedback, which has helped to improve the manuscript, and we are pleased that our revision has addressed the reviewer's requests.

Reviewer #5

The manuscript entitled "X-linked deletion of Crossfirre, Firre, and Dxz4 in vivo uncovers diverse phenotypes and combinatorial effects on autosomal gene regulation" by Hasenbein et al. follows on the authors previous work on Firre and Dxz4 and presents a study of mice harbouring combinations of X chromosomal mutations in Firre, Dxz4, and Crossfirre, which is an antisense transcript that partially overlaps Firre. All genotypes are viable and the triple mutation can be bred in homozygous state suggesting subtle effect on development. This is consistent with the authors earlier work on homozygous Firre and Dxz4 double mutations in mice but also extends by including Crossfirre.

The authors have expectations that any of these genes could be regulating X inactivation in redundant manner but an analysis of X-linked allelic gene expression does not find evidence for this hypothesis. Notably, effects on autosomal gene expression is observed suggesting a role in gene regulation. The authors go on to perform an unbiased phenotyping using a service. Although, several phenotypes are recorded, the overall connection and conclusion remains a bit unclear and it could be made more clear what is biologically relevance as the mice seem overall healthy.

The manuscript stands out for the large amount of data. Although some constitutes negative evidence an advance is made over the authors earlier analysis that did not include Crossfirre (both Super Loup and Megadomain formation were already abrogated by Firre/Dxz4 mutations in the authors previous work.) The relevance of the results can likely be increased by adding to the text to the effect that the view of Crossfirre as a major regulator of imprinting and XCI might need correction.

Comments:

1. Fig 6. A focus of the relevant phenotyping data would make the study accessible to a wider readership. The current data presentation will not be useful for a wider readership and the expert would take advantage of seeing the histology analysis and other types of experimental data from the various assay). This would allow the authors also to include targeted analysis of certain aspects. In particular FACS analysis of the spleen populations. The present Fig 6 is certainly very useful way to summarize for the database and best be included as supplement along with the statistical view.

We sincerely thank the reviewer for the suggestion to include detailed results of the mouse phenotyping in the manuscript. We would like to mention that we intentionally chose to present a high-level overview of all findings in order to provide a complete picture of the TKO phenotyping screen (Figure 6). However, we agree that the detailed results of the individual tests, particularly for histology and FACS analysis, are of great interest. Thus, the complete set of raw measurements from the relevant phenotyping tests, including histology and FACS, are made available on the Phenomap website of the German Mouse Clinic (GMC, <https://www.mouseclinic.de>).

We clarified this matter in the Results section of the manuscript [line number 331] and in the Data availability section: “A complete set of raw measurements from the relevant phenotyping tests is available on the Phenomap website of the GMC (<https://www.mouseclinic.de>).”

This webpage includes a section with all GMC phenotyping results (<https://tools.mouseclinic.de/phenomap/jsp/annotation/public/phenomap.jsf>, as depicted below).

Legend:
■ Clear genotype related differences ■ Subtle findings ■ No significant differences ■ Not analyzed ■ Data still in preparation Report Links to detailed phenotyping data (please click)

Option 1: Search for a single gene
 Find projects by gene symbol:

Option 2: Load all data (it might take a while!)

Gene: **Gm35612 (Crossfire), Firre, 4933407K13Rik (Dx24)** (1 of 1) 25 1

Mutant project	Gene	Mutation type	Zyosity	Bone & Cartilage	Behaviour	Neurology	Eye & Vision	Nocturnop	Energy Metabolism	Clinical Chemistry	Immunology	Allergy	Steroid Metabolism	Cardio-vascular	Lung Function	Expression Profiling	Pathology	Bodyweight	Report
Crossfire-Firre-Dx24_TKO	Gm35612 (Crossfire), Firre, 4933407K13Rik (Dx24)	Genome editing	homo, homo	■	■	■	■	■	■	■	■	■	■	■	■	■	■	■	■

Graphic: Clinical Chemistry: Lactat (AU400) boxplot with stripchart, split by sex and genotype

Table caption: FACS Blood Leukocytes: Percentage of leukocytes in blood plasma. Medians, first and third quartile and p-values calculated by a Wilcoxon rank-sum test.

	female		male		p-value	p-value	p-value
	con n=15	homo n=13	con n=15	homo n=13			
Percent granulocytes	11.9 [9.56, 15.5]	12.3 [2.56, 7.50]	12.3 [2.56, 7.50]	13.8 [11.6, 17]	0.955	0.1	0.427
Percent neutrophils	10.3 [9.56, 12.9]	9.48 [8.3, 11]	10.6 [8.465, 12.2]	11.4 [10, 15.2]	0.3	0.145	0.699
Percent eosinophils	2.11 [1.665, 2.545]	2.79 [2.49, 3.16]	1.82 [1.53, 2.215]	1.73 [1.41, 2.07]	0.007	0.058	0.12
Percent B cells	50.5 [48.6, 53.25]	48.1 [46.6, 50.6]	58.4 [56.3, 62.4]	56.1 [50.5, 57.8]	0.188	0.03	0.028
Percent T cells	24.7 [23.5, 25.6]	24.6 [21.9, 26.2]	16.8 [15.3, 18.25]	16.6 [15.4, 17.8]	0.865	0.812	0.88
Percent monocytes	6.90 [5.54, 7.84]	8.32 [7.83, 8.99]	7.26 [5.57, 7.965]	8.65 [7.13, 10.8]	0.003	0.029	< 0.001
Percent NK cells	3.57 [3.035, 4.15]	3.99 [3.32, 4.2]	3.49 [3, 3.885]	3.37 [3.18, 3.74]	0.345	0.751	0.571

Media caption: Photomicrographs of hematoxylin and eosin-stained heart sections of a control mouse and a homozygous mouse showing a dilated ventricle or a vessel defect (red arrows). Right panel shows the photomicrograph of the heart removed from the same homozygous mouse at the time of necropsy (15 weeks) showing a dilated wall (black arrow).

Media caption: Photomicrographs of hematoxylin and eosin-stained lung sections of a control mouse and a homozygous mouse showing multilocalized congestive arteries (arrows) with thickening of the vascular wall with an inflammatory cell surrounding (enlarged panel).

To specifically view our project's raw measurement data, please enter one of the three genes knocked out in our study and select the autocomplete suggestion. This will forward you to the repository storing all phenotyping results of the TKO mice, including histology images and details of the FACS analysis. Please click on the individual phenotyping category to access the results of the particular phenotyping tests.

2. line 164: Fig 2d has a table from which it might appear that the litter size of the TKO is increased (TKO 9.4, delta Crossfirre 6.3, delta Crossfirre and Firre 6.9, wild type missing). In the context that TKO display subtle phenotypes, the increased litter size might be noteworthy. Is this statistically significant? If the TKO mice are fitter, is there any indication that the DXZ4 / Firre system might be a parasitic genetic system?

We thank the reviewer for bringing the increased TKO litter size to our attention. We have now increased the number of analyzed litters to 10 for each KO strain and observed comparable litter sizes (updated Figure 2d, see below). Additionally, we observed no differences in sex ratios. Based on these findings, there is no evidence to suggest that TKO mice are fitter than the other genotypes. The updated table is provided below for your convenience.

d

Mating GT [-/- x -/y]	Litter	Pups	Litter size	♂	♀	p-value	Sex ratio
Δ Crossfirre	10	64	6.4	33	31	0.45	106.5
Δ Crossfirre-Firre	10	62	6.2	32	30	0.45	106.7
TKO	10	67	6.7	34	33	0.5	103.0

Fig. 2: Mice carrying a *Crossfirre* single deletion or combined with *Firre* and *Dxz4* are viable and undergo normal development.

d, Sex distribution of homozygous Δ Crossfirre, Δ Crossfirre-Firre, and TKO breeding. The p-values are obtained from a binomial test, assuming an equal number of peaks between the sexes.

3. The effect of the mutations on allelic expression suggests that imprinted and random XCI are comparable to controls. The authors use this to argue against a function in XCI. However, subtle effects on stability of chromatin or repression on the Xi might be missed. From Ciz1 mutant mice and conditional Xist mutation in mice would be blood cell hyperplasia (splenomegaly) or tumours at old age. It would be important to include a statement if aged mice were analysed and such blood cell issues ruled out. This would strengthen the idea of dispensability for maintenance of Xi repression.

We thank the reviewer for the suggestion and acknowledge the interest in studying aged mice lacking *Crossfirre*, *Firre*, and *Dxz4*. We understand that the analysis of such a model could provide important insights, particularly in detecting effects on the maintenance of Xi repression, which may lead to conditions such as blood cell hyperplasia or tumor development during aging. While we agree that assessing age-related phenotypes could further strengthen our conclusions, our current study was focused on investigating the effects of the TKO in the embryonic and adult stages. However, it is an intriguing experiment that we may explore in the future.

To clarify that we cannot rule out TKO-related phenotypes during aging, we have included the following statement in the Discussion [line number 363]: “However, we cannot rule out potential effects beyond the adult stage we investigated, particularly during aging.”

Minor points

1. *Crossfirre* is described by the authors as an imprinted gene which often affect embryonic growth in a subtle manner. It would be interesting to investigate if there were a growth effect during transient embryonic growth period, which often can be compensated as development progresses towards birth.

We thank the reviewer for this insightful suggestion. While we did not perform a detailed investigation of embryonic growth at multiple stages, we did isolate E12.5 embryos and placentas and observed no noticeable developmental defects at this stage. Additionally, our transcriptomic analysis of the placentas revealed very few differentially expressed genes between WT and TKO (Figure 3), further supporting the absence of major developmental abnormalities at this stage. While it would be interesting to explore *Crossfirre*'s effects at different stages of development, this falls beyond the scope of our current study.

2. The authors show that different autosomal genes are shown to be misregulated by (cross) *firre* and *DXZ4*. Conversely, has an analysis been performed to of the ncRNAs affect autosomal genes via miR regulation (sponge)? Or are these misregulated genes related to transposable or mobile genetic elements?

We appreciate the reviewer's insightful comments and agree that the idea of *Crossfirre*, *Firre* and *Dxz4* controlling autosomal gene regulation via miRNA sponging or piRNA biology is indeed an exciting possibility. To address these potential mechanisms, we conducted the following analyses:

1. miRNA sponging hypothesis: To investigate whether *Crossfirre*, *Firre* or *Dxz4* regulate autosomal genes by acting as miRNA sponges, we utilized the three major miRNA target prediction databases (miRBase, TargetScan and miRDB). Our analysis revealed no miRNA binding sites in any of the three lncRNAs, suggesting that miRNA sponging is unlikely to be the mechanism by which these lncRNAs affect autosomal gene regulation.
2. Transposable element hypothesis: We interpret the reviewer's suggestion to explore whether piRNAs produced by these lncRNA loci could explain the regulation of autosomal genes, particularly through piRNA-mediated regulation of transposable elements. To address this question, we investigated whether the 73 differentially expressed autosomal genes, shared by TKO and Δ *Crossfirre-Firre* in the spleen, are direct targets of piRNAs. We cross-referenced these genes with the piRNA target database piRBase and found no evidence that they are direct piRNAs targets.

In conclusion, while miRNA sponging and piRNA-mediated regulation are intriguing possibilities, our current findings suggest that neither mechanism fully explains the regulation of these autosomal genes by *Crossfirre*, *Firre*, and *Dxz4*. However, given the complexity of ncRNA-mediated gene regulation, we cannot completely rule out these pathways at this stage. Further comprehensive studies, including more in-depth analysis of potential indirect interactions via small RNA pathways, are required to fully exclude these hypotheses.

We sincerely thank the reviewers for their time and effort in evaluating our manuscript. Their insightful feedback was valuable. We have addressed the comments and provide point-by-point responses below. We believe that the revisions have substantially improved the manuscript.

Reviewer #5

The authors have addressed all of my comments in their comprehensive response. In particular, the availability of the large amount of phenotyping data is an important asset of the study, which will be of high interest to researchers in noncoding RNA, gene regulation and mouse development.

We thank the reviewer for the positive feedback and are pleased that our revisions have met the expectations.